# Insights into the Currently Available Drugs and Investigational Compounds Against RSV with a Focus on Their Drug-Resistance Profiles

**DOI:** 10.3390/v17060793

**Published:** 2025-05-30

**Authors:** Alessia Magnapera, Anna Riccio, Antonio Curcio, Caterina Tramontozzi, Lorenzo Piermatteo, Stefano D’Anna, Stefano Alcaro, Claudia Alteri, Simone La Frazia, Anna Artese, Romina Salpini, Valentina Svicher

**Affiliations:** 1Department of Biology, University of Rome Tor Vergata, 00133 Rome, Italy; anna.riccio@uniroma2.it (A.R.); caterina.tramontozzi@alumni.uniroma2.eu (C.T.); lorenzo.piermatteo@uniroma2.it (L.P.); stefano.danna@alumni.uniroma2.eu (S.D.); simone.lafrazia@gmail.com (S.L.F.); r.salpini@gmail.com (R.S.); 2Department of Health Science, Magna Graecia University, 88100 Catanzaro, Italy; antonio.curcio@unicz.it (A.C.); alcaro@unicz.it (S.A.); artese@unicz.it (A.A.); 3Department of Oncology and Hemato-Oncology, University of Milan, 20122 Milan, Italy; claudia.alteri@unimi.it; 4Microbiology and Virology Unit, Fondazione IRCCS Ca’ Granda Ospedale Maggiore Policlinico, 20122 Milan, Italy

**Keywords:** RSV, novel drugs, mutations, drug resistance, structural dynamics

## Abstract

Respiratory syncytial virus (RSV) is a leading cause of severe respiratory illness in infants, young children, as well as elderly and immunocompromised patients worldwide. RSV is classified into two major subtypes, RSV-A and RSV-B, and remains the most frequently detected pathogen in infants hospitalized with acute respiratory infections. Recent advances have brought both passive and active immunization strategies, including FDA-approved vaccines for older adults and pregnant women and new monoclonal antibodies (mAbs) for infant protection. Although significant progress has been made, the need remains for improved antiviral treatments, particularly for vulnerable infants and immunocompromised patients. Recent studies have identified multiple RSV mutations that confer resistance to current treatments. These mutations, detected in both in vitro studies and clinical isolates, often complicate therapeutic outcomes, underscoring the need for updated and effective management strategies. In this context, evaluating protein flexibility through tools like DisoMine provides insight into how specific mutations impact structural dynamics at binding sites, thus affecting ligand affinity. This review aims to synthesize these aspects, offering a comprehensive insight into ongoing efforts to counteract RSV and address the evolving challenge of drug resistance.

## 1. Introduction

Human respiratory syncytial virus (HRSV) was first identified in chimpanzees with upper respiratory tract illness in 1956 and, in the following year, was isolated from infants with severe lower respiratory tract illness [1,2]. RSV is classified into two major antigenic subtypes, RSV-A and RSV-B, based on the genetic characteristics of the region encoding the G protein, co-circulating with the same frequency [3]. Over decades, RSV has become widely recognized as a leading cause of respiratory infections that disproportionally affect young children worldwide [4,5]. RSV contributes to millions of lower respiratory tract infections (LRTIs) each year, resulting in millions of hospitalizations and many fatalities [6]. In 2019, in particular, RSV-related lower respiratory tract infections accounted for 338,495 deaths globally, particularly among children and older adults. Effective treatments and preventive options have been limited, underscoring the urgent need for safe and effective approaches to improve RSV management [7]. The development of RSV vaccines was hindered for decades due to safety concerns and adverse events from a vaccine trial conducted in the 1960s, with a formalin-inactivated RSV vaccine that caused enhanced respiratory disease (ERD) syndrome, resulting in serious illness among children [8,9]. Progress in RSV prevention has accelerated with the development of both passive and active immunization options. Particularly in recent years, promising advances in vaccine development have led to a range of promising candidates, including vector-based, particle-based, subunit, mRNA, live-attenuated, and chimeric vaccines [10]. Among RSV proteins, the F protein is the most studied target due to its critical role in viral entry and fusion with host cells. Its importance in the viral life cycle makes it a key target for vaccine and drug development. To date, the Food and Drug Administration (FDA) has approved two subunit vaccines, RSVPreF3 (Arexvy, GSK, May 2023) and RSVpreF (Abrysvo, Pfizer, July 2023), for the prevention of LRTIs caused by RSV for usage in adults older than 60 years [11,12]. The RSVpreF vaccine was also approved for pregnant women as a single dose at 32 to 36 weeks of gestation for the prevention of RSV-related LTRI in infants under 6 months old [13]. In addition, the FDA also approved the first mRNA vaccine (m-RNA-1345-mRESVIA, Moderna) which encodes the stabilized RSV pre-fusion F glycoprotein for the prevention of RSV in adults over 60 years of age [14]. Regarding passive immunization in infants, two monoclonal antibodies (mAbs) that target different sites on the F protein are available. Palivizumab (Synagis, AstraZeneca), the first mAb approved in 1998, is a humanized monoclonal IgG antibody capable of specifically inhibiting an epitope at the A antigenic site of the F protein (this protein has a 92% homology between RSV subtypes A and B). This mAb is used as a prophylactic strategy in high-risk children [15,16]. In July 2023, the FDA approved Nirsevimab (Beyfortus, Sanofi and AstraZeneca), a long-acting mAb, for the prevention of RSV in infants. The CDC’s Advisory Committee on Immunization Practices (ACIP) recommended Nirsevimab for infants younger than 8 months of age during their first RSV season (October–March) and for high-risk children aged 8 to 19 months who are entering their second RSV season [17,18].

In spite of the advancements in preventing RSV infection, there is still a significant need for improved antiviral treatments, specifically for infants and immunocompromised patients, who are at higher risk of severe disease and for whom vaccination may not be sufficient. Ribavirin, a nucleoside analogue, is the only FDA-approved antiviral treatment for RSV, but it has limited efficacy and high cost [19,20]. Over time, different anti-RSV agents, including small molecules, have been developed. These compounds are in preclinical evaluation or have entered clinical trials and, as they target different RSV proteins, they are considered very promising for the treatment of RSV infection [21,22,23].

Several mutations that affect the activity and potency of anti-RSV drugs (both standard and new) have been identified in both clinical isolates and in vitro studies, representing a primary obstacle to the effectiveness of the therapy. This study aims to review the existing literature on the role of mutations known to alter the efficacy of the currently available anti-RSV drugs. Furthermore, we also use structural analysis to evaluate the ability of single amino acids, at positions associated with drug resistance, to affect the flexibility of the protein structure and, in turn, drug susceptibility, thus providing a potential structural explanation for the role of mutations in drug-resistance emergence.

## 2. Epidemiology and RSV Genotype Distribution

RSV represents a global health threat, as its burden is associated with acute LRTIs in infants and young children, as well as in elderly and immunocompromised patients [24,25]. RSV infection is one of the major causes of childhood morbidity and mortality, especially in low- and middle-income countries. According to the World Health Organization (WHO) global RSV surveillance program, initiated in 2019, the estimated incidence was 33.1 million cases of RSV-associated acute LRTIs, causing 3.6 million hospitalizations and more than 100,000 in-hospital deaths among children younger than 5 years [4,26]. The severity of RSV infections varies largely, ranging from mild cold-like symptoms to more severe lower respiratory tract complications, like bronchiolitis and pneumonia, particularly in fragile populations [27]. In particular, children under 5 years of age infected with RSV are the most likely to be hospitalized every year, with 60–70% of children infected before 1 year of age [28]. Among adults, the disease is mostly mild to moderate, but it can lead to hospitalization, serious complications, and death in older individuals and those with impaired immunity [29]. In 2019, approximately 5.2 million cases, 470,000 hospitalizations, and 33,000 in-hospital deaths were reported in ≥60-year-old adults in high-income countries [30]. In Europe, RSV causes severe respiratory tract infections, with seasonality correlating with latitude. Geographical spreading of the infection shows variations in RSV season length, with longer seasons in northern European countries than in the United States [31]. It is interesting to note that the COVID-19 pandemic has influenced the epidemiology of RSV and its usual seasonality. In particular, in the United States, RSV epidemics in the three years preceding the COVID-19 pandemic exhibited a consistent October-to-December onset, lasting around 27 weeks. Conversely, after the COVID-19 pandemic, the RSV epidemic of 2021–2022 began 21 weeks earlier, reaching its peak in July and lasting 33 weeks, with an RSV-positive percentage comparable to that of pre-pandemic seasons. During the 2022–23 year, the epidemic lasted 32 weeks, with onset occurring in June and with a greater positivity than the pre-pandemic seasons [32]. Similar shifts in the 2021–2022 seasonal patterns were observed in other European countries [33,34]. As for the 2022–2023 RSV seasonality, data suggest a gradual return to the seasonal patterns observed in pre-pandemic years [34]. Different theories have been proposed as potential explanations for this evidence: SARS-CoV-2 infections may have induced immune dysregulation in children, increasing their susceptibility to RSV; the viral interactions between SARS-CoV-2 and RSV may have caused an increase in viral co-infections or superinfections; or the prolonged lack of RSV circulation early in the pandemic may have contributed to a decreased viral immunity in vulnerable age groups [35].

RSV can be divided into two antigenic groups: A and B, with multiple genotypes within each group. There are 9 genotypes identified for RSV-A and at least 32 genotypes for RSV-B, according to the genetic profiles observed in RSV gene G [36,37]. The prevalence of RSV-A and RSV-B can vary from year to year and by geographic location: The two genotypes circulate globally, and their prevalence can fluctuate in different regions and seasons. From a clinical perspective, RSV-A cases are associated with increased hospitalisation rates and greater disease severity, suggesting a potential higher virulence than RSV-B [38].

## 3. RSV Virion and Genome

RSV is an enveloped, single-stranded, negative-sense RNA virus that belongs to the order of *Mononegavirales*, family *Pneumoviridae*, genus *Orthopneumovirus* [39]. The RSV genome is 15.2 kb and contains 10 genes, encoding 9 structural and 2 nonstructural proteins: non-structural protein 1 (NS1) and 2 (NS2), nucleoprotein (N), phosphoprotein (P), matrix protein (M), small hydrophobic protein (SH), attachment protein (G), fusion protein (F), the cofactor proteins M2-1 (a transcription processivity and anti-termination factor) and M2-2 (a polymerase cofactor responsible for the switch between transcription and replication), and the polymerase protein (L) [39,40,41,42] (Figure 1).

When visualised by electron microscopy, RSV virions are either spherical particles with a diameter of 100–350 nm or long filaments up to 10 μm in length and 60–200 nm in diameter [42,43]. The RSV virion comprises a ribonucleoprotein complex (RNP) composed by the viral genome bound to four proteins: the nucleoprotein (N), the large polymerase protein (L), the phosphoprotein (P), and the processivity factor M2-1. This structure protects the RNA and forms the minimal replication machinery. The matrix protein (M) is located between the RNP and the envelope, acting as a bridge. The RSV envelope contains three integral membrane proteins: the receptor attachment glycoprotein (G), the fusion protein (F), and a small hydrophobic (SH) protein. The SH protein, being present with less abundance in the envelope, acts as an ion channel when activated at low pH. Although its role in viral replication is still undefined, several studies suggest that SH protein may play a role in RSV pathogenicity and immune evasion [39,44,45]. The two major surface glycoproteins, the G and the F proteins, are essential for viral infectivity. In particular, the protein G protrudes from the viral envelope and acts as the primary factor for RSV attachment to host cell receptors, such as glycosaminoglycans (GAGs), heparan sulfate proteoglycans (HSPGs), and fractalkine receptor CX3C-chemokine receptor 1 (CX3CR1) [46,47,48,49,50,51,52]. On the other hand, the F protein is the key mediator for fusion between the viral envelope and the host cell membrane. When the F protein is activated, it shifts from the metastable pre-fusion conformation to a stable post-fusion conformation, leading to the fusion between the viral and cellular membranes [53,54].

Importantly, the F and G proteins have distinct yet complementary roles in both viral entry and immune evasion. The G glycoprotein facilitates initial viral attachment through interactions with host receptors, including CX3CR1, and contains a conserved CX3C motif that mimics the host chemokine fractalkine. This mimicry disrupts leukocyte chemotaxis and reduces immune cell recruitment, contributing to immune evasion [55,56]. Additionally, its heavily glycosylated mucin-like domains help mask neutralising epitopes through glycan shielding [57]. In contrast, the F protein is directly responsible for membrane fusion and is activated by receptor-mediated triggers or pH changes, undergoing major conformational rearrangements to mediate viral entry [58]. Mutations in the F protein and conformational transitions from the pre-fusion to post-fusion state can also hinder recognition by neutralising antibodies, especially those targeting pre-fusion specific epitopes [59]. The interplay between G and F proteins enhances viral entry efficiency and helps RSV evade host immune surveillance.

Like the spikes of other respiratory viruses, the G and F proteins are major targets of host-neutralising antibodies and may be key pharmacological targets for small molecules that impair their function at several steps, from attachment to the host cell to intracellular trafficking and viral morphogenesis [39,60,61,62,63]. The RSV genome is a single-stranded, negative-sense RNA of 15,191–15,226 nucleotides. At the 5′ and 3′ ends, both the cap and the polyA tail are absent and genes are transcribed in the order 3′–NS1-NS2-N-P-M-SH-G-F-M2-L–5′ into ten independent messenger RNAs (mRNAs). Each mRNA encodes a single major protein, with the exception of M2, which has two slightly overlapping ORFs, coding for M2-1 and M2-2 proteins [43,64]. The 3′ and 5′ ends, upstream of the NS1 gene and downstream of the L gene, contain two extragenic regions 44 nt and 155 nt long, respectively. Each gene is preceded by a relatively conserved region called the gene-start (GS) region, 9 nt long, and terminates with a 12–14 nt gene-end (GE) signal, also fairly conserved, ending in a 4–7 U tail. The first nine genes are separated by intergenic regions of variable length (1–58 nt). The last two genes, M2 and L, overlap by 68 nt [43]. Protein M2-1, 194-amino acids long, is a regulator of transcription acting as an anti-terminator factor that can be found in both glycosylated and non-glycosylated forms and generates a homotetramer via a localised oligomerization domain on residues 32–63 [40]. This protein possesses a zinc-finger motif that can bind viral RNAs, and these interactions are critical to the M2-1 protein’s ability to support RNA synthesis [65,66]. NS1 and NS2 proteins, 139 and 124 amino acids long, respectively, have a nonstructural function and are supposed to interfere with the innate immune response by inhibiting the induction of IFN-α/β [67,68]. These proteins also suppress premature apoptosis through the activation of NF-κB, thus increasing cell survival and, in turn, promoting viral replication [69].

## 4. Replication Cycle

The RSV entry into the host cell is the outcome of the cooperative role of the G and F proteins, which ensures a well-orchestrated interplay between attachment and fusion. The F protein mediates fusion between the viral envelope and host cell membranes once activated by the binding of the G protein to the receptor [70,71].

Upon G protein attachment, the F protein undergoes conformational changes, shifting from a pre-fusion to a post-fusion state. This change is essential for membrane fusion, allowing the virus to enter host cells [53] (Figure 2).

It should be noted that mutant viruses lacking the G gene were also found to infect cells, presumably by attachment through heparin binding domains on the F protein [74]. During the fusion process between the cell membranes and the viral envelope, the fusion peptide is embedded in the host cell membrane, while heptad repeat 1 and 2 (HR1 and HR2) regions undertake a conformational change to facilitate the fusion, forming a six-helix bundle structure and bringing viral envelope and cellular membranes closer to each other [54]. Following attachment and fusion, the RNP complex is released into the cytoplasm of the cell [75]. The RNP complex is replicated, transcribed, and translated into viral proteins needed for the assembly of new infectious virions in a cytoplasmic space called the inclusion body (IB) [76]. The RNA-dependent RNA polymerase (RdRp) acts as a transcriptase, beginning to cap, transcribe, and polyadenylate viral mRNAs, as well as non-proofreading polymerase to replicate the genome, producing positive-sense RNA intermediates (antigenomes) that become the template for generating genomic RNA for packaging into virions [77]. Although the mechanism for switching between transcription and replication is not fully elucidated, the N and the M2-2 proteins are reported to be significant in this regulatory process [41]. During their synthesis, both genomic and antigenomic RNAs are encapsulated by N proteins [78]. RSV glycoproteins are first translated into the endoplasmic reticulum and then matured into the Golgi apparatus, where they are terminally glycosylated, and they are finally transferred to the plasma membrane in regions rich with lipid rafts. The RNP complex linked to the RSV RNA genome is transported to the plasma membrane, where the assembly and budding of viral particles take place [79]. It is noteworthy that the F protein of RSV alone is capable of initiating the process of fusion, without the need for any other additional viral component, leading to the formation of syncytia. This type of structure (visible as the presence of multinucleated cells) is a distinctive feature of RSV infection, and it consists of the merging of infected cells with adjacent uninfected cells (fusion from within), thus acting as a mechanism to promote viral spread through the target cells [39].

## 5. RSV Proteins as Targets of RSV Inhibitors and Related Drug-Resistance Mutations

The following paragraphs are dedicated to the description of RSV proteins that are targets of the currently available anti-RSV drugs and investigational compounds. Furthermore, this section will also review the mutations associated with drug resistance, covering both those confirmed to confer resistance in in vitro studies and those observed in clinical studies, which may be involved in resistance but need further validation. For those mutations characterised in in vitro experiments, the degree of resistance observed for each mutant towards a compound is expressed by the ratio EC_50-mut_/EC_50-wt_, known as fold change resistance (FC). EC_50_ is the concentration of the inhibitor that causes a 50% reduction in viral yield. In this article, FC values > 1000 were considered to confer high levels of resistance to treatment, FC values ranging from 1000 to 100 were considered intermediate resistance, and FC < 100 were considered low resistance. FC values below 1 were considered to confer hypersusceptibility to treatment, meaning the drug potentially exhibits enhanced efficacy in the presence of specific mutations, resulting in improved therapeutic outcomes. It is noteworthy that the majority of published research on compounds that are effective against RSV is based on “laboratory strains.” In particular, the RSV strain A2, recognised as the prototype A strain for RSV research, was first identified in 1961 from the lower respiratory tract of an infant in Melbourne, Australia [80]. Conversely, the RSV strain B9320 is widely used in in vitro experiments as a prototype B genotype [81,82]. Table 1 provides a detailed description of specific amino acid substitutions across different RSV proteins and their impact on drug resistance in terms of FC values.

In order to provide a comprehensive characterisation of drug-resistance mutations, the following paragraphs also provide a structural characterisation of these mutations in terms of their localisation in the three-dimensional structure of RSV proteins and impact on the flexibility of the protein. These analyses were performed using A2, B and B9320 as backbone references.

## 6. Materials and Methods

### 6.1. Search Strategy and Eligibility Criteria

A search in the international NCBI database was conducted by two study team members. The following keywords were used: RSV, mutation, drug resistance, drug names, as well as in vitro experiments [83]. Articles published within the time window ranging from 2010 to January 2025 were included. In vitro studies were considered if they had been performed at least in duplicate using standardised procedures. Only studies published in English were included. The exclusion criteria were as follows: (i) Reviews, letters, personal opinions, book chapters, case reports, conference abstracts, and abstract-only publications were excluded.

### 6.2. Structural Analysis

All protein structures were obtained from the RCSB Protein Data Bank (RCSB PDB) [84], while the drug molecules were analysed using data from PubChem [85], using the Simplified Molecular Input Line Entry System (SMILES) for each compound to generate the corresponding two-dimensional structures. Maestro graphical interface and UCSF ChimeraX were used to generate images of the three-dimensional structure of the analysed RSV proteins and the two-dimensional structures of the drugs listed in Table 1 [86,87].

The influence of the analysed drug-resistance mutations on protein flexibility was assessed using DisoMine, a machine learning-based tool specifically designed to predict protein disorder by analysing both backbone and side chain dynamics based only on the protein sequence. DisoMine employs a neural network trained on experimental disorder data to estimate the degree of flexibility at each residue position [88]. In this review, the protein sequences of both wild-type (WT) and mutants of the fusion glycoprotein, nucleoprotein, major surface glycoprotein, and polymerase of RSV were submitted to DisoMine. The side chain dynamics disorder scores were obtained between WT and mutant proteins to evaluate whether the introduced mutations significantly affected the protein’s intrinsic flexibility. For backbone dynamics, values above 0.8 indicate rigid conformations, values above 1.0 correspond to membrane-spanning regions, and values below 0.69 suggest flexible regions. Scores ranging between 0.69 and 0.80 are considered context-dependent, meaning they can exhibit either rigidity or flexibility depending on their structural environment. Similarly, regarding side-chain dynamics, higher values suggest a greater probability of rigidity.

## 7. Fusion Glycoprotein (F)

The RSV fusion glycoprotein (F) is a type I fusion protein of 574 amino acids, experiencing a significant conformational change from a trimeric, metastable pre-fusion state to a highly stable post-fusion structure [89]. During its biosynthesis, the F protein assembles into a homotrimer and undergoes cleavage at two adjacent furin cleavage sites. This proteolytic processing of the precursor protein (F0) results in the generation of two polypeptides, F1 and F2, which remain linked by a disulphide bond. The F1 subunit has a predicted N-glycosylation site at N500, while the F2 subunit reveals two N-glycosylation sites at N27 and N70 [90,91]. Upon cleavage, the F1 subunit, which contains the fusion peptide at its N-terminus, is exposed. The fusion peptide is a hydrophobic, glycine-rich segment that plays a critical role in the fusion process by inserting into the target cell membrane and facilitating viral entry [47,92]. Furthermore, in vitro studies have shown that RSV with a deletion in the G protein gene can replicate in the absence of the G protein by using the F protein for both attachment and fusion [53]. Although the G protein is not essential for viral replication, an F and G protein complex has been observed to assemble on the infected cell surface. During this assembly, the G protein plays the role of stabilising the F protein in the pre-fusion conformation, preventing premature membrane fusion that would compromise virus exit from the infected cell and consequently decrease its infectivity [71]. This multifaceted role of the F protein in RSV pathogenesis underscores its importance as a target for therapeutic intervention. The F protein is also a major antigenic target for neutralising antibodies due to its higher immunogenicity and greater conservation compared to the RSV G protein. Consequently, many current research efforts focus on stabilising the pre-fusion form of the F protein, which is recognised as the most immunogenic structure [89,93] (Figure 3).

Inhibiting the F protein can be targeted at several stages: from the F1 fusion peptide, which initiates fusion, to the final stage when the post-fusion protein irreversibly refolds into a six-helix bundle complex, culminating in membrane fusion and the formation of a stable fusion pore [94]. So far, inhibiting the F protein and the viral fusion can be achieved by using approved mAbs or by targeting the viral protein with promising small molecules.

### 7.1. Monoclonal Antibodies

Since the F protein is a surface protein, it is highly antigenic and can be targeted therapeutically using mAbs [95]. Palivizumab, a recombinant humanised monoclonal immunoglobulin, was approved in 1998 for prophylaxis against severe RSV disease in selected high-risk groups. It effectively binds to the RSV fusion protein, inhibiting subsequent viral infection [96]. This binding occurs on site II of the F protein, a helix-turn-helix motif present on both the pre-fusion (pre-F) and post-fusion (post-F) conformations [97]. Approximately 50% of the pre-F and post-F protein surfaces overlap one another, including antigenic sites II and IV [59].

Palivizumab prophylaxis reduced RSV-related hospitalizations by 55%, with an even greater reduction of 78% in premature children without bronchopulmonary dysplasia (BPD). In children with BPD, there was a 39% reduction in hospitalizations [98]. The treatment also decreased the severity of RSV outcomes, including fewer intensive care unit (ICU) admissions and hospitalization days. Another mAb is Nirsevimab, a long-acting intramuscular recombinant neutralising human IgG1κ mAb targeting the prefusion conformation of the RSV F protein. Nirsevimab has been modified with a triple amino acid substitution in the Fc region to extend its serum half-life. Several mutations in F proteins may have implications for resistance to these (Table 1).

**Table 1 viruses-17-00793-t001:** List of mutations in RSV proteins implicated in drug resistance.

Protein	Drug Class	Drug Name	Amino Acid Substitution	Domain	Strain ^a^	Fold Change Resistance Value ^b^	Reference
**F**	Monoclonal	Palivizumab	S275L ^c^	Antigenic site A	A2	>25,000	[99]
	Antibodies		S275F ^c^	Antigenic site A	A2	>25,000	[99]
			K272E ^c^	Antigenic site A	B	>25,000	[99]
			K272M ^c^	Antigenic site A	B	>25,000	[99]
			K272T ^c^	Antigenic site A	A2	>25,000	[99]
			K272N ^c^	Antigenic site A	B	5164	[99]
			K272Q ^d^	Antigenic site A	A2	>3000	[100]
			N262Y ^d^	Antigenic site A	A2	512	[100]
			T400A ^c^	Cysteine-Rich	A2	3.3	[101]
			L138F ^d^	Fusion Peptide	A2	2.4	[100]
			K399I ^d^	Cysteine-Rich	A2	2.3	[100]
			F488L ^d^	Heptad Repeat 2	B	1.4	[100]
			F488S ^d^	Heptad Repeat 2	B	1.1	[100]
			F488Y ^d^	Heptad Repeat 2	A2	1.1	[100]
			E487D ^d^	Heptad Repeat 2	A2	1	[100]
			D486N ^d^	Heptad Repeat 2	A2	0.8	[100]
			F140L/N517I ^d^	F1	A2	0.6	[100]
			N262D ^d^	Antigenic site A	A2	NA	[102]
			S276N ^d^	Antigenic site A	B	NA	[103]
		Nirsevimab	N208D ^c^	F1	B9320	>90,000	[82]
		(MEDI8897)	K68N/N208S ^c^	F1/F2	B9320	>90,000	[82]
			N208S ^c^	F1	B9320	24,618	[82]
			K68N/N201S ^c^	F1/F2	B9320	13,438	[82]
			N67I/N208Y ^c^	F1/F2	A2	102.5	[82]
			N201S ^c^	Heptad Repeat 1	B9320	64.5	[82]
			K68N ^c^	F2	B9320	3.8	[82]
			N67I ^c^	F2	A2	1.5	[82]
			N208Y ^c^	F1	A2	1.1	[82]
	Small Molecules	Presatovir	F488L ^d^	Heptad Repeat 2	B	>5000	[100]
		(GS-5806)	F488S ^d^	Heptad Repeat 2	B	>5000	[100]
		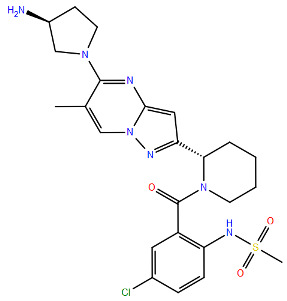	L138F ^d^	Fusion Peptide	A2	>2000	[100]
			F140L/N517I ^d^	F1	A2	>2000	[100]
			D486N ^d^	Heptad Repeat 2	A2	1193	[100]
			T400I ^d^	Cysteine-Rich	A2	410	[101]
			T400A ^d^	Cysteine-Rich	A2	214	[100]
			K399I ^d^	Cysteine-Rich	A2	87	[100]
			F488Y ^d^	Heptad Repeat 2	A2	75	[100]
			E487D ^d^	Heptad Repeat 2	A2	35	[100]
			K394R ^c^	Cysteine-Rich	A2	4	[104]
			K272Q ^d^	Antigenic site A	A2	2	[100]
			N262Y ^d^	Antigenic site A	A2	0.2	[100]
		Rilematovir	K394R ^c^	Cysteine-Rich	A2	6024	[100]
		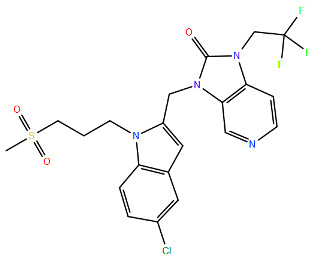					
		VP-14637	T400A ^d^	Cysteine-Rich	A2	>3200	[100]
		(MDT-637)	D486N ^d^	Heptad Repeat 2	A2	>3200	[100]
		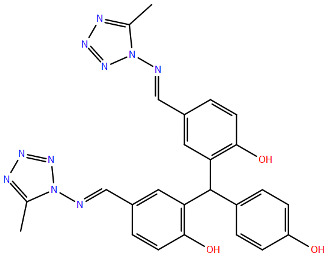	F488L ^d^	Heptad Repeat 2	B	>2500	[100]
			F488S ^d^	Heptad Repeat 2	B	>2500	[100]
			L138F ^d^	Fusion Peptide	A2	>2000	[100]
			F140L/N517I ^d^	F1	A2	>2000	[100]
			E487D ^d^	Heptad Repeat 2	A2	75	[100]
			F488Y ^d^	Heptad Repeat 2	A2	52	[100]
			K399I ^d^	Cysteine-Rich	A2	45	[100]
			N262Y ^d^	Antigenic site A	A2	0.6	[100]
			K272Q ^d^	Antigenic site A	A2	0.6	[100]
		BMS-433771	K394R ^c^	Cysteine-Rich	A2	1902	[104]
		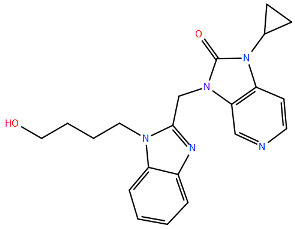	L141F ^c^	Fusion Peptide	A2	>18.3	[105]
		TP0591816	L141F ^c^	Fusion Peptide	A2	>4720	[105]
		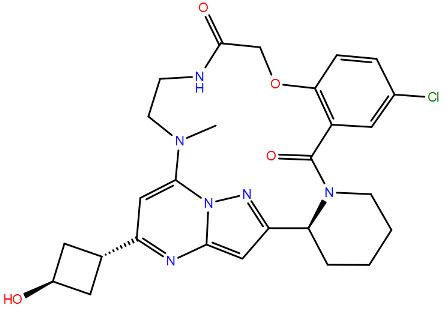					
		TMC-353121	K394R ^c^	Cysteine-Rich	A2	1033	[104]
		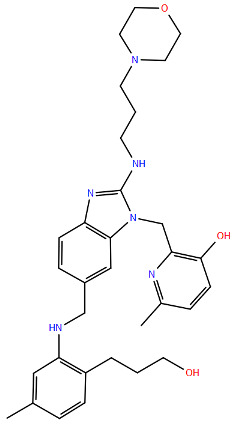					
		Ziresovir	K394R ^c^	Cysteine-Rich	A2	355	[104]
		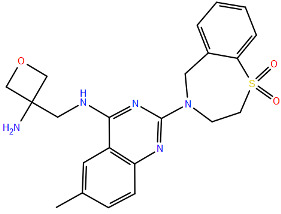					
		LF-6	K394R ^c^	Cysteine-Rich	A2	>71	[104]
		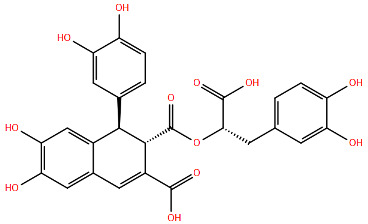					
**N**	Small Molecules	Zelicapavir	Q102L/M109T/I129M ^c^		A2	42.4	[106]
		(EDP-938)	L139Q ^c^		B	42	[106]
		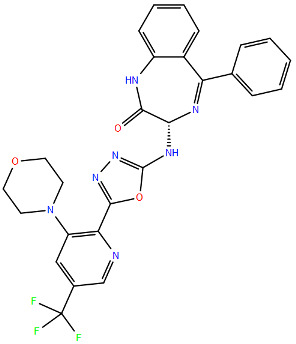	M109K ^c^		A2	26.9	[106]
			M109T ^c^		A2	5.4	[106]
			I129M ^c^		A2	3.7	[106]
			T29S/S134T ^c^		A2	3.3	[106]
			K136R ^c^		A2	2.7	[106]
			S134T ^c^		A2	2.2	[106]
			Q102L ^c^		A2	2	[106]
			M109I ^c^		A2	1.6	[106]
**G**	Small Molecules	Zelicapavir	K205G/K213G/T219A ^c^		A2	60	[106]
		(EDP-938)	R8H ^c^		A2	3.1	[106]
**L**	Nucleoside	ALS-8112	A789V ^f^	Motif B	A2	Km = 61.0 ± 26.9 μM	[107]
	Analogue	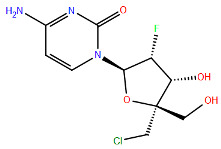	M628L ^f^	Motif F	A2	Km = 23.1 ± 0.07 μM	[107]
			I796V ^f^	Motif B	A2	Km = 22.0 ± 9.9 μM	[107]
			L795I ^f^	Motif B	A2	Km = 16.5 ± 6.4 μM	[107]
	NN inhibitor	AZ-27	Y1631H ^e^	Capping domain	A2	940	[107]
		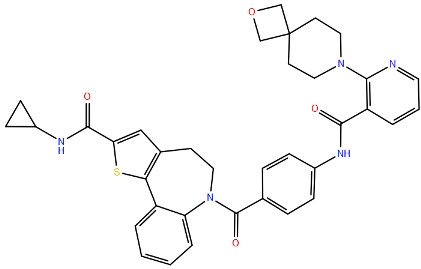					

^a^ The listed RSV mutations were studied in a laboratory setting using the standard reference strains A2, B, and B9320. ^b^ The level of resistance conferred by each mutant to a specific compound is measured as the fold change resistance (FC), expressed by the ratio EC_50-mut_/EC_50-wt_. EC_50_ is the concentration of the inhibitor that causes a 50% reduction in viral yield. ^c^ In vitro studies. ^d^ Clinical isolate. ^e^ Enzymatic assay. ^f^ Enzymatic assay conducted with the nucleotide analogue 4′ClCH_2_-CTP. All two-dimensional drug structures were generated using the Maestro graphical interface [86].

These mutations in the antigenic sites of F proteins can disrupt the binding of mAbs, thus reducing their neutralization activity and leading to therapeutic resistance [108]. These mutations are predominantly located within the antigenic site A, the fusion peptide, and the heptad repeats [104,109] (Table 1 and Figure 4).

#### 7.1.1. Mutations Associated with Resistance to Palivizumab

The S275L substitution in antigenic site A of the F protein is known to confer high-level resistance to Palivizumab, exhibiting a FC > 25,000 [99], without affecting the efficiency of viral replication in in vitro experiments [110] (Table 1). High-level resistance has also been observed for S275F. The replacement of serine with phenylalanine or leucine at this position likely causes substantial steric hindrance, preventing effective binding of the antibody to the F protein [111].

Similarly, substitutions at residue 272 of the F protein play a critical role in mediating resistance to Palivizumab. In particular, clinical isolates with K272E and K272M in the RSV-B strain and K272T in the RSV-A2 strain are known to confer high-level resistance with a FC > 25,000 [99]. High-level resistance is also observed in presence of K272Q (FC > 3000) in the RSV-A2 strain and K272N (FC > 5000) in the RSV-B strain. These mutations were predominantly observed in viruses collected from individuals who had been exposed to Palivizumab, suggesting a treatment-induced selective pressure [99,100]. The K272Q variant, although structurally similar, does not show the same level of resistance as K272E, indicating that even small changes in the amino acid sequence can lead to significant differences in the interaction between the virus and the drug [100]. This variability in resistance suggests that the substitution of lysine (K) with glutamine (Q) at position 272 may alter the binding affinity of Palivizumab, but the exact level of resistance may depend on additional factors such as the presence of other mutations or structural variations in the F protein [99,111].

In the RSV-A2 strain, the N262Y substitution in antigenic site A confers a FC of 512, resulting in a significant reduction in susceptibility to Palivizumab [100]. The importance of asparagine (N) at this position is further demonstrated by several binding kinetics studies, which show the effect of the N262D substitution conferring resistance to Palivizumab [102,111].

Conversely, other mutations have been identified to confer extremely low resistance (FC < 5). This is the case for T400A substitution located in the cysteine-rich region of the F1 subunit in the RSV-A2 strain (FC 3.3) (Table 1). This low-level resistance presumably indicates that this substitution does not entirely disrupt the interaction between Palivizumab and the F protein [101]. Similarly, the L138F substitution in RSV-A2 strains leads to a FC of 2.4 [100]. The leucine residue at position 138 plays a supportive role in maintaining the structural integrity of the binding site, and its replacement with phenylalanine moderately reduces the binding efficiency of the antibody [100]. The K399I substitution in the RSV-A2 strain has been associated with a resistance FC of 2.3 [100]. A FC < 1.5 has been observed for the substitutions at positions 287 and 288, located in the heptad repeats, suggesting minimal contributions of these substitutions to drug resistance.

Interestingly, substitutions D486N and F140L/N517I in the RSV-A2 strain are found to confer potential hypersusceptibility to Palivizumab with FCs of 0.8 and 0.6, respectively [100].

Zhu et al. reported that the N276S substitution in RSV-A and the S276N substitution in RSV-B did not confer resistance to Palivizumab, as confirmed by neutralization assays using both clinical isolates and recombinant virus [102]. However, experiments in murine models revealed that altering the amino acid in position 276 in the F protein led to the emergence of monoclonal antibody-resistant mutants (MARMs) [103].

#### 7.1.2. Mutations Associated with Resistance to Nirsevimab

Different mutations in the F1/F2 subunits of the F protein have been shown to confer resistance to Nirsevimab in in vitro experiments (Table 1). In particular, two substitutions at position 208 (N208D or N208S) in the RSV-B9320 strain confer high levels of resistance to Nirsevimab, with FCs > 90,000 and 20,000, respectively [82]. Notably, the degree of resistance conferred by N208S is dramatically increased when this substitution is co-present with K68N (FC > 90,000). Similarly, K68N/N201S double substitutions in the RSV-B9320 strain show a high level of resistance (FC > 13,000), whereas the single N201S shows a level of resistance of 64.5-fold. Notably, K68N and N201S seem to play a role in Nirsevimab resistance mainly in the RSV strain B9320. Indeed, a previous study based on in vitro selection experiments showed the emergence of these mutations only in the RSV strain B9320 under the selective pressure imposed by Nirsevimab [82]. This is an interesting finding, highlighting that the genetic backbones of the different genotypes may predispose them to the generation of genotype-dependent genetic profiles associated with drug resistance.

In the RSV-A2 strain, intermediate resistance (>100-fold) was observed in variants carrying the double substitution N67I/N208Y but not in variants harbouring N208Y or N67I alone [82].

### 7.2. F Protein Inhibitors

This section discusses the small-molecule RSV fusion inhibitors under investigation and the related drug-resistance profiles.

#### 7.2.1. TMC-353121 and BMS-433771 and Their Resistance Profiles

TMC-353121 and BMS-433771 bind with high affinity to a three-fold-symmetric pocket within the central cavity of the pre-fusion RSV F. These inhibitors interact via aromatic stacking with two residues of phenylalanine (F140 and F488) in two labile regions crucial for membrane fusion, thus acting as antagonists that stabilise the metastable prefusion conformation of RSV F. Additionally, the binding of TMC-353121 involves the formation of a distorted 6HB bundle, with TMC-353121 sandwiched between HR1 and HR2 [90]. Mutations, conferring resistance to these drugs, occur in residues directly involved in binding with the inhibitors or are involved in the conformational flexibility required to accommodate inhibitor binding. However, viruses harbouring these mutations do not propagate as efficiently as wild-type RSV in vitro, suggesting a remarkable decrease in viral fitness [112]. Thus, it is plausible to hypothesise that viral strains carrying such mutations can rarely emerge in vivo as a consequence of their impaired viral replicative capacity.

Regarding BMS-433771, substitutions such as L141F and K394R in the F protein have been associated with resistance to this drug in the RSV-A2 strain. The FCs associated with these substitutions indicate a variable decrease in BMS-433771 potency against RSV, ranging from to >18.3 for L141F to 1902 for K394R [104,105] (Table 1).

#### 7.2.2. Presatovir and Its Resistance Profiles

Another important inhibitor is Presatovir (GS-5806), an oral RSV fusion inhibitor with potent and selective anti-RSV activity in vitro. In a human challenge study of healthy volunteers, Presatovir significantly reduced the RSV viral load and the severity of clinical disease [113]. This compound is predicted to adopt a conformation similar to TMC-353121, occupying all three lobes of the binding pocket and interacting with F140 and F488 through aromatic stacking of its phenyl and pyrazolo[1,5-a]pyrimidine groups [112].

Several mutations in the F protein have been identified to confer resistance to Presatovir (Table 1). F488L and F488S substitutions in the RSV-B strain lead to a significant decrease in Presatovir potency against RSV, with a FC > 5000 [100]. The L138F substitution, found in clinical isolates of RSV-A2, results in resistance to this drug, with a FC > 2000 [100]. Additionally, the F140L/N517I double substitution in the F protein of RSV-A2 clinical isolates also confers resistance to Presatovir, with a FC similar to L138F [100]. Moreover, D486N substitution in the RSV-A2 strain was found to confer a high level of resistance to Presatovir, with a FC > 1000 [100]. Similarly, the T400I substitution in the RSV-A2 strain shows a substantial resistance level of 410-fold [101]. Moderate levels of resistance are observed with other substitutions in the RSV-A2 strain, including a T400A substitution that exhibits a FC of 214 and K399I with a value of 87. F488Y and E487D in the RSV-A2 strain show moderate resistance levels, as well, with values of 75 and 35, respectively [100]. The K394R substitution in the RSV-A2 strain shows a resistance value greater than 4, indicating a low level of resistance compared to other substitutions [104]. Furthermore, the K272Q substitution in RSV-A2 shows low resistance, with a FC of 2, while N262Y substitution, with a FC of 0.2, confers a potential hypersusceptibility to Presatovir [100].

#### 7.2.3. Rilematovir (JNJ-53718678) and Ziresovir (RO-0529, AK0529) and Their Resistance Profiles

Rilematovir and Ziresovir are orally bioavailable RSV inhibitors that have demonstrated efficacy in Phase 2 studies, with the former tested in a challenge study involving healthy adults. Both are currently being evaluated in hospitalised infants and adults. These compounds effectively inhibit RSV F protein-mediated fusion with the host cell membrane, preventing viral entry and demonstrating good safety profiles [114,115]. In particular, K394R was identified in RSV-A2 in in vitro studies with a FC of 6024, whereas no resistance value has been observed for the other substitutions [104] (Table 1). Recently, Song and colleagues optimised a caffeoylquinic acid derivative, CL-A3-7, able to reduce both wild-type and K394R variant RSV infections in vitro and in vivo [116].

#### 7.2.4. VP-14637 and Its Resistance Profiles

VP-14637 is a small molecule inhibitor targeting the RSV F protein that seems to be a promising treatment against RVS infection; however, the emergence of resistance mutations presents a significant challenge to its clinical efficacy [117]. In particular, the T400A substitution, located in the cysteine-rich F1 region of the A2 strain, confers a very high resistance, with a value > 3200, making this one of the strongest resistance mutations in the dataset, alongside D486N, sited in the heptad repeat region, which also shows resistance greater than 3200 [100] (Table 1). The F488L and F488S substitutions in the heptad repeat region were found in the RSV-B strain, and they confer a high level of resistance (FC > 2500), while the F488Y substitution in the RSV-A2 strain shows a much lower resistance of 52-fold. Both substitutions L138F and F140L/N517I in the RSV-A2 strain exhibit high levels of resistance, with FC values exceeding 2000. Substitutions like E487D in the heptad repeat region and K399I in the cysteine-rich region of the F protein (RSV-A2 strain) confer a lower level of resistance with FC values of 75 and 45, respectively, indicating a partial reduction in the drug’s efficacy. On the other hand, the N262Y and K272Q substitutions in antigenic site A (RSV-A2 strain) confer potential hypersusceptibility to this molecule, with a FC of 0.6 [100].

#### 7.2.5. TP0591816 and Its Resistance Profiles

TP0591816 is another small molecule targeting the RSV F protein. Yoshida and collaborators optimised macrocyclic pyrazolo[1,5-a]pyrimidine derivatives to obtain TP0591816, a potent inhibitor of the F protein that hinders RSV-induced cell–cell fusion. This drug was first synthetised to be active against resistant RSV strains that possessed D486N, F488L, or T400I substitutions, showing good results. So far, the substitution L141F was the only substitution associated with a reduced sensitivity to this specific compound [105]. The substitution L141F observed in the RSV-A2 strain disrupts key interactions between the compound and the F protein, thereby impairing its antiviral activity. The FC >4000 associated with this mutation indicates a significant decrease in TP0591816 potency against RSV [105] (Table 1).

#### 7.2.6. Salvianolic Acid R (LF-6) and Its Resistance Profiles

Salvianolic acid R (LF-6) is another promising compound acting as an RSV fusion inhibitor, isolated from *Mesona chinensis Benth*. This compound is able to suppress RSV infection, but resistant variants for LF-6 harbouring the K394R substitution in the viral F protein were identified. This substitution also confers resistance to other fusion inhibitors, thus acting as a cross-resistance mutation [86] (Table 1).

### 7.3. Multi-Drug Resistance Mutations

The K394R substitution found in the RSV-A2 strain F protein significantly impacts RSV resistance to multiple fusion inhibitors, and for this reason, it can be defined as a multi-drug resistance mutation (Table 1). In particular, a previous study showed that the presence of K394R alone or in combination with D486N or D489Y can enhance the membrane fusion activity of the F protein, as well as decrease the proportion of pre-fusion F on the cell surface, thus resulting in a shorter time window for inhibitors to bind [104]. This can represent a novel mechanism conferring cross-resistance to multiple fusion inhibitors.

### 7.4. Structural Characterisation of Drug-Resistance Mutations

The next step was to evaluate the ability of the above-mentioned drug-resistance mutations to affect the flexibility of the protein structure, thereby contributing to the impact on the binding capacity of particular inhibitors. Indeed, it is well-documented in the literature that mutations in specific amino acid residues of well-known protein targets can alter the protein’s flexibility and dynamic behaviour, thereby modifying the binding of a specific drug to its target. In some cases, these alterations lead to drug resistance, which may be associated with either increased flexibility or enhanced rigidity [118,119,120,121,122]. This analysis was performed by applying DisoMine, a tool capable of predicting the disordered state of a protein, in terms of its backbone and side chains, using only its sequence [88] (Table 2, Table 3 and Table 4). Our purpose was to conduct a preliminary analysis of how such mutations might alter the flexibility of specific amino acid positions in key RSV protein targets, laying the groundwork for more accurate and detailed future investigations. Indeed, such analyses are necessary to determine whether increased flexibility or rigidity of an amino acid residue and, consequently, of the entire protein can be linked to a lower binding affinity and an enhanced drug resistance. Regarding the backbone dynamics, values above 0.8 indicate rigid conformations, values above 1.0 indicate membrane-spanning regions, and values below 0.69 indicate flexible regions. Values between 0.69 and 0.80 are ‘context’ dependent and capable of being either rigid or flexible. These values are highly dependent on the amino acid type [123].

For the fusion glycoprotein, the impact of mutations on protein flexibility and corresponding rigidity at specific mutated sites was assessed across the three genotypes [88] (Table 2, Table 3 and Table 4). Based on this analysis, we can conclude that mutations in genotypes B and B9320 result in varied structural outcomes: Approximately 50% of cases exhibit increased rigidity, while the other 50% demonstrate enhanced flexibility. Conversely, genotype A2 consistently shows a pronounced increase in rigidity regarding the backbone.

In genotype A2, a significant increase in backbone rigidity was observed for the following amino acid substitutions: N208Y (+75), N67I (+80), N262Y (+85), S275L/F (+89 and +90), and N517I (+80). Notably, the substitutions at position 275 are associated with high resistance to Palivizumab, while the substitution at position 517 correlates with consistent resistance to Presatovir. In genotype B, the most significant change was observed with the F488S substitution, which led to a notable increase in backbone flexibility (−89). Interestingly, this substitution is associated with elevated resistance to Presatovir and VP-14637. Moreover, we described several mutations found in RSV-resistant strains that are still not associated with specific values of resistance to Palivizumab compared to the wild-type strain (N262D and S276N). These mutations could potentially play a role in inhibiting drug activity, and they require further investigation.

## 8. Nucleoprotein (N)

The RSV nucleoprotein (N) is organised into two globular domains, the N-terminal domain (NTD, amino acids 31–252) and C-terminal domain (CTD, amino acids 253–360), with flexible arms at the N- and C-termini (aa 1–30 and 362–391) [124]. The protein plays a pivotal role by tightly binding to the viral RNA genome, thereby forming a ribonucleoprotein (RNP) complex [65]. RNP is the template for transcription and replication by the RdRp complex, which consists of the polymerase (L) and the phosphoprotein (P), whose interaction with RNP triggers viral replication [44,125]. Initially, when viral RNA is absent or present in limited quantities, the nucleoprotein N predominantly exists in a monomeric form. This form is kept soluble and non-aggregated through interaction with the phosphoprotein P. This protein acts as a chaperone, preventing the premature self-association of N monomers [124]. When viral RNA is synthesised, the N monomers quickly associate with the RNA to form the homodecameric RNP complex. The transition from monomer to homodecamer occurs in a coordinated manner with the synthesis of viral RNA. The phosphoprotein P facilitates this process by helping to load N onto the viral RNA. Once the RNA is coated by N in its homodecameric form, P mediates the interaction of the RNP complex with the polymerase L [126] (Figure 5).

The nucleoprotein/phosphoprotein interaction site is crucial for both the production of new virions and the effectiveness of antiviral drugs designed to inhibit viral replication by targeting this nucleoprotein interface. Specifically, the phosphoprotein interaction areas are located between the αI2 and αN1 helices in the nucleoprotein N-NTD. In particular, the last two residues of the phosphoprotein, F241 and D240, interact with R132, Y135, R150, and H151 of the nucleoprotein [127].

### N Protein Inhibitors

To date, six classes of N inhibitors have been evaluated: one peptide, one siRNA, and four small molecules. Among these small molecules, Zelicapavir (EDP-938) and RSV-604 are characterised by the same dihydrobenzodiazepinone scaffold [128].

Substitutions in the Q102-L139 region confer similar degrees of resistance to Zelicapavir and RSV-604. These drugs are characterised by a similar chemotype and bind to the N central core, thus affecting N–N and N–viral RNA interactions [128] (Figure 6).

In RSV-A2 genotype, the Q102L/M109T/I129M triple substitution shows a FC of 42.4, while the single M109T, I129M, and Q102L substitutions are characterised by resistance factors of 5.4, 3.7, and 2, respectively [106] (Table 1). The L139Q substitution (RSV-B strain) and M109K substitution (RSV-A2 strain) result in 42-fold and 26.9-fold increases, respectively, indicating moderate resistance levels. The double T29S/S134T mutant has a 3.3-fold increase resistance to Zelicapavir, while the single S134T substitution shows a 2.2-fold increase (Table 1). Overall, although single mutations confer resistance, their impact is typically more pronounced when in combination with other mutations [106].

Also, for the N protein, the effect of amino acids at a specific position associated with drug resistance on protein flexibility and corresponding rigidity at specific mutated sites was evaluated across the two genotypes [88] (Table 5 and Table 6).

For the A2 genotype, the resistance mutation involving the substitutions of glutamine with leucine at position 102 and the isoleucine with methionine at position 129 are among those that have shown the most pronounced variation in backbone flexibility, at +43 and −40, respectively. When combined, they are associated with the highest levels of resistance to Zelicapavir.

## 9. Major Surface Glycoprotein (G)

The major surface glycoprotein (G) is a type II, highly glycosylated membrane protein consisting of 292–319 amino acids. It includes an intracellular cytoplasmic tail (aa 1–37), a transmembrane domain (aa 38–66), and an extracellular domain extending to the carboxy terminus [129]. Its central region contains a conserved CX3C chemokine motif, which lacks glycosylation and acts as a chemokine mimic [58]. This central domain is flanked by two large, highly variable mucin-like domains, making it the most variable RSV protein [130]. Excluding the central region, G is a highly glycosylated protein. While glycosylation is not essential, it appears to enhance the surface expression of the G protein, leading to initial contact with the host cells [131]. It has been proven that the central conserved domain (CCD) mediates binding to CX3CR1 in primary human airway epithelial cells in vitro, serving as the first step in cellular infection. Additionally, the anti-G antibodies effectively neutralize RSV in these cells [56,132]. Moreover, RSV can bind to immobilised heparin. In vivo, heparin is mainly found in mast cells and basophil granules, while its related compound, heparan sulphate, is present on most mammalian cell surfaces and in the extracellular matrix. Many viruses use heparan sulphate to attach to and infect target cells. Thus, heparin-binding proteins interact with heparin through electrostatic interactions between the negatively charged sulphate groups on heparin and the positively charged amino acids in the protein’s heparin-binding domain (HBD). Notably, the ectodomain of the RSV-G protein contains a cluster of positively charged amino acids (180P→K233) within an immunodominant region [133]. A complete crystallographic structure of the G protein is not available. The most complete and high-resolution structure to date is PDB ID 5WNA, which includes a significant portion of the CCD. This structure features the binding site for a high-affinity, broadly neutralising human mAb [109] (Figure 7). Moreover, there is also a secreted form of the G protein that acts as a neutralization decoy and a modulator of leukocytes with Fc gamma receptors, helping RSV immune escape [134].

### G Protein Inhibitors

Among RSV proteins, G is characterised by the lowest degree of genetic conservation [43]. In the setting of therapeutic prophylaxis, mAbs targeting the G protein have been associated with a better prognosis as a consequence of reduced viral load [130]. In certain cases, anti-G protein mAbs have proven to be more effective than anti-F protein mAbs by reducing airway inflammation in mouse models [135]. By targeting the CX3C motif specifically, these anti-G protein mAbs not only neutralise RSV but also mitigate disease-related symptoms, as observed by reduced bronchoalveolar lavage (BAL) cell infiltration, improved Th1/Th2-cytokine balance, and better lung pathology in mice [136,137,138]. Additionally, Fc-mediated neutralization has been confirmed for anti-G CX3C mAbs, which are also capable of modulating IFN responses. A recent co-crystallisation of two broadly neutralising human mAbs, 3D3 and 2D10, revealed that these antibodies bind to unique epitopes within the RSV G protein’s central conserved domain, neutralise RSV in vitro with complement, and inhibit G protein CX3C-CX3CR1 chemotaxis [109].

So far, different mutations have been described in the G protein mainly as a result of genetic drift, rather than conferring resistance to drugs. Interestingly, some substitutions in the G protein have been shown to increase the level of resistance to N inhibitors when present along with N substitutions. The triple mutant K205G/K213G/T219A in G protein, when co-expressed in recombinant viruses with the triple mutant Q102L/M109T/I129M in N protein, confers a level of resistance to Zelicapavir of 60-fold. The single substitution R8H in the G protein, when in association with substitution M109I in the N protein, shows a reduced susceptibility to Zelicapavir, albeit at a low level [106] (Table 1). It is conceivable to hypothesise that Zelicapavir either is directly targeting the N protein or is exerting its antiviral activity through the N protein via another, still unidentified, factor. Overall, based on current literature, mutations in the G protein can increase the degree of resistance when present along with mutations in the N protein, thus acting as secondary mutations. As recently highlighted for other viruses such as HIV [139], this finding highlights that the genetic profile’s underlying resistance to a specific drug class can be more complex than currently thought, thus involving interactions among multiple viral proteins. Further studies are necessary to better clarify the mechanisms underlying this phenomenon.

The DisoMine tool was adopted to evaluate G glycoprotein flexibility in both the wild-type (WT) and mutated sequences [88] (Table 7).

## 10. Polymerase (L)

The L protein, comprising 2165 amino acids, contains three essential enzymatic domains: the RNA-dependent RNA polymerase (RdRp) domain, the polyribonucleotidyl transferase (PRNTase or capping) domain, and the methyltransferase (MTase) domain, which is responsible for cap methylation [140]. The C-terminal domain (CTD) is located after the MTase domain and is recognised as the most variable region among non-segmented negative-sense RNA viruses. Although its primary sequence shows a low degree of genetic conservation, it is suggested that this domain, enriched in basic amino acids, forms a clamp with the MTase domain, facilitating RNA substrate recognition for methylation [141]. Structurally, the N-terminal RdRp domain of the RSV L protein adopts a right-handed architecture, a common feature of all viral RNA-dependent RNA polymerases. This domain is organised into four subdomains: palm, fingers, thumb, and an additional supportive subdomain. The RSV RdRp is characterised by three conserved regions (CR_I, CR_II, and CR_III) and six conserved sequence motifs (motifs A–F), with the majority residing in the palm subdomain. This subdomain comprises two α-helices and a β-sheet consisting of five strands, with these motifs playing crucial roles in accommodating the RNA template, facilitating nucleotide incorporation, and ensuring polymerase flexibility [142].

The phosphoprotein (P) interaction site has a pivotal role in the L protein’s polymerase function. P, a tetrameric protein, acts as a crucial polymerase cofactor by linking L to the nucleoprotein-RNA complex and modulating RNA synthesis through its interactions with L, RNA-free N (N0), N, and M2-1. Despite this, the detailed interactions between L and P, and the precise RNA synthesis mechanism employed by the RSV polymerase, are not yet fully understood [143]. The recent developments of 3D models of the L protein bound to the P tetramer have provided valuable insights into the role and mechanism of the RSV RdRp complex [140,143,144] (Figure 8).

### L Protein Inhibitors

Inhibiting RSV polymerase activity is a promising strategy for discovering anti-RSV antivirals. Specifically, several nucleoside analogue inhibitors (NIs) and non-nucleoside inhibitors (NNIs) have been identified, and some of them, such as JNJ-8003, interact with the capping domain [144]. The capping domain of a viral polymerase plays a crucial role in the replication of RNA viruses. Its primary function is to add a protective cap structure to the 5′ end of the newly synthesised viral RNA, a process known as capping, which is useful for shielding the viral RNA from degradation by exonucleases [145]. At this domain, substitutions at position 1631 confer resistance to various inhibitors, including AZ-27, a potent benzothienoazepine derivative of YM-53403. AZ-27 is about 75 times more effective than YM-53403, with an EC_50_ of 10 nM against RSV strain A2 under multicycle growth conditions [146] (Figure 8). As an RdRp inhibitor, AZ-27 targets both mRNA transcription and RNA replication in RSV, inhibiting transcription initiation at the 3′ end of the template more than at the 5′ end. Importantly, AZ-27 does not disrupt the L–P interaction, which is crucial for viral RNA synthesis. The compound effectively blocks de novo RNA synthesis and elongation of back-primed RNA, though it does not inhibit the addition of 1 to 3 nucleotides by back-priming. Resistance to AZ-27 arises from mutations at position 1631 of the L protein, enabling the RdRp to overcome the transcription initiation block imposed by the ligand [146,147].

Another promising polymerase inhibitor is Lumicitabine (ALS-008176 or ALS-8176), a prodrug of the nucleoside analogue ALS-8112, currently in Phase II development for RSV treatment (Figure 8). After oral administration, Lumicitabine is metabolised to the cytidine nucleoside analogue ALS-8112, which circulates in the bloodstream and is then converted intracellularly to its antiviral-active 5′-nucleoside triphosphate (NTP) [148]. The efficacy of Lumicitabine is affected by mutations in key regions of the RSV polymerase, particularly in motifs B and C.

Motif B, characterised by a conserved glycine-rich sequence (GGxxG), is the tract where most drug-resistance mutations, known as “QUAD” substitutions, occur. These mutations enhance the RSV RdRp ability to discriminate between natural NTPs and NTP analogues used as inhibitors [149]. Motif C (CR_III) contains the catalytic GDN sequence (G810, D811, N812), while a catalytic aspartate residue (D700) is located in motif A. These residues are crucial for coordinating the two magnesium ions required for catalysing phosphodiester bond formation, essential for RNA synthesis [150] (Figure 9).

The QUAD substitutions (M628L, A789V, L795I, I796V) are associated with in vitro resistance to ALS-8112; three of these substitutions are located within motif B (A789V, L795I, I796V), while the last one is within motif F [107,140]. The presence of four substitutions in the L gene leads to a 39-fold decrease in RSV inhibition potency by ALS-8112. The use of its 5′-triphosphate metabolite (ALS-8112-TP) in the presence of the QUAD substitutions results in a 145-fold decrease in inhibition potency compared to the wild-type protein [107]. It has been shown that ALS-8112-TP inhibits the polymerase activity of the RSV L-P RNA polymerase complex by causing an immediate termination of chain synthesis [107].

To evaluate the role of individual substitutions, experiments were conducted with 4′ClCH_2_-CTP, a nucleotide analogue already known to cause a stronger resistance phenotype conferred by the QUAD substitutions than ALS-8112-TP [107,151]. Enzymatic assays reveal that the Km values were comparable between the WT (19.7 ± 6.7 μM), M628L (23.1 ± 0.07 μM), L795I (16.5 ± 6.4 μM), and I796V (22.0 ± 9.9 μM) variants, indicating that these three mutations have no significant impact on the recognition of the 4′ClCH_2_ moiety. Conversely, the A789V substitution resulted in a moderate increase in the Km (61.0 ± 26.9 μM), though it was lower than the Km observed with the QUAD mutant (235 ± 77 μM) [151].

The Y1631H substitution in the RSV-A2 genotype shows a resistance FC of 940, indicating a high level of resistance to AZ-27. AZ-27 targets the capping domain of the L-protein, leading to significant reduction in the drug’s efficacy, as the capping domain is crucial for mRNA synthesis and stability and mutations in this region can severely disrupt antiviral action [151].

The effects of these specific mutations on residues backbone rigidity and flexibility were assessed using DisoMine [88] (Table 8).

The most significant change is observed in the Y1631H substitution, which is associated with a substantial increase in FC to the AZ-27 inhibitor.

## 11. Discussion

RSV is a major public health concern, causing acute respiratory tract infections worldwide, with particularly high disease impact on paediatric populations, older adults, and immunocompromised individuals. It is a leading cause of childhood morbidity and mortality, with over 3.6 million hospitalizations and nearly 100,000 in-hospital deaths among children under five years of age each year [4,26]. Given the significant global burden of RSV, the development of effective pharmacological strategies to inhibit viral replication is pivotal and still represents an unmet medical need.

This review has provided an extensive overview of the currently available anti-RSV compounds and their related drug-resistance profiles. The current knowledge of drug-resistance profiles is mainly derived from in vitro experiments in which laboratory RSV strains (mainly the A2 strain) have been used to infect different cell cultures. A large spectrum of drug-resistance mutations has been described, conferring different degrees of drug resistance to anti-RSV drugs. The fact that most studies are based on in vitro experiments can represent a limitation, as the type and the extent of drug resistance can vary according to the viral strain used and the experimental conditions. This variability arises from multiple factors, including selective pressures on the in vitro system and the possibility of enhanced viral evolution under controlled conditions. Conversely, in an in vivo setting, other factors can influence virus mutation selection, such as immune system competence, competition with other pathogens, and drug distribution within different tissues. Consequently, the mutations that confer drug resistance in vitro may not always emerge or predominate in vivo. On this basis, further studies are necessary to fully characterise drug-resistance profiles emerging in patients. In keeping with this concept, another parameter that should be elucidated is the impact of resistance mutations on viral replication. This issue needs to be better explored because drug-resistance mutations are detrimental for viral fitness and tend not to be selected in vivo.

Another issue that deserves further attention is represented by mutations conferring multi-drug resistance, further complicating treatment strategies and limiting therapeutic options. In this setting, the K394R mutation is of particular interest, since it plays a crucial role in multidrug resistance, potentially compromising the efficacy of several F inhibitors [104,116].

As highlighted in this review, RSV presents numerous potential protein targets, each with distinct functions and structural features. Therefore, gaining a comprehensive understanding of these key viral proteins is a crucial step toward the rational design and discovery of potent, selective inhibitors capable of overcoming issues related to viral mutations. In this context, it is crucial to gain a deeper understanding of how these substitutions influence the structure of the RSV key proteins, particularly near specific targetable sites, and their impact on the binding affinity of known ligands and inhibitors. Thus, in silico techniques, ranging from molecular modelling to machine learning approaches, can be extremely useful. The structural analysis investigated whether resistance-associated mutations might affect the flexibility of viral proteins, thus providing a mechanistic hypothesis for resistance development. In this review, the DisoMine tool allowed us to perform a preliminary, though limited, analysis of protein flexibility; however, more accurate and in-depth studies are required to fully elucidate how these substitutions alter protein behaviour, potentially leading to drug resistance. It was observed that key substitutions, particularly in the RSV A2 F protein (e.g., S275L/F, N262Y, N517I), led to substantial increases in backbone rigidity. These findings are consistent with the idea that modifications in protein flexibility at specific sites can affect ligand accessibility or binding affinity, potentially reducing antiviral effectiveness [152]. Interestingly, in RSV genotype B strains, mutations such as F488S were associated with increased flexibility and high resistance to Presatovir and VP-14637, suggesting that both excessive flexibility and rigidity can negatively impact drug binding, depending on the ligand’s binding mechanism.

Nevertheless, this study has important limitations, since the DisoMine tool provides sequence-based predictions without incorporating full 3D structural context or explicit ligand interactions. Thus, while informative, these results should be considered as a basis for hypothesis generation, requiring validation through molecular dynamics (MD) simulations or experimental techniques such as NMR and cryo-EM.

Overall, so far, large amounts of data have been generated on resistance profiles to anti-RSV drugs. On this basis, the future direction should imply a better understanding of such profiles an in vivo setting. It is pivotal to optimise the use of these drugs in terms of proper starting times and durations for anti-RSV treatment and frequency of virological monitoring. On the other hand, an adequate detection of drug-resistance profiles in a clinical setting should also imply the need for robust genotypic resistance testing for prompt and accurate characterisations of drug-resistance profiles. At the same time, there is the need to deepen the impact of drug-resistance mutations on the structural conformation of the viral protein. This information is crucial for the design of new drugs that are also effective against drug-resistant strains.

In conclusion, these findings could play a pivotal role in developing new, potent antiviral drugs against RSV, contributing to overcoming challenges associated with drug resistance. Continuing efforts to extend our knowledge of the RSV proteome is essential for handling the challenges associated with therapeutic resistance. Such efforts will enhance our ability to determine how specific amino acid substitutions can alter protein structural features, as well as influence drug binding and efficacy.

## 12. Conclusions

RSV continues to impose a substantial burden on global health, affecting vulnerable populations and straining healthcare systems.

Despite advances in RSV treatment, no cure exists for RSV infection, driving research into small molecules targeting RSV proteins, especially the F protein. The emergence of drug resistance presents a substantial challenge to the effectiveness of both existing and experimental therapies for RSV. RSV inhibitors, including neutralising antibodies and small molecules, undergoing preclinical and clinical trials are associated with a high risk of selecting resistant mutants due to the high mutation rates of the viral genome. Investigating the role of mutations that emerge in vivo is crucial for identifying potential resistance mechanisms, predicting viral adaptation to therapies, and developing more effective and targeted treatments that can overcome resistance. Overall, the data reported in this review highlight the importance of strengthening ongoing research on RSV in order to understand viral evolution and enhance treatment options, thereby reducing the burden of RSV-related diseases.

## Figures and Tables

**Figure 1 viruses-17-00793-f001:**
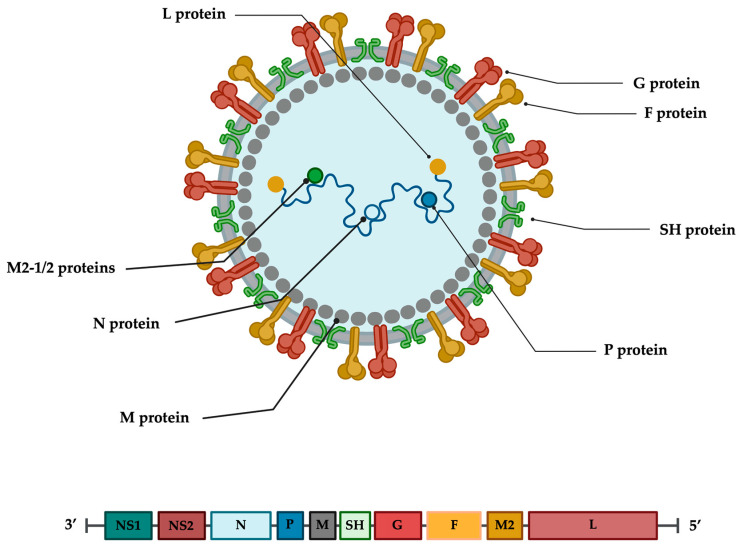
**Schematic representation of the RSV structure and genome**. RSV is a negative-sense, single-stranded RNA virus that encodes eleven proteins: nine structural and two non-structural. The non-structural proteins are NS1 and NS2. The viral envelope contains the fusion glycoprotein (F), the attachment glycoprotein (G), and the small hydrophobic protein (SH). The matrix protein (M) is located on the inner surface of the viral envelope. Four nucleocapsid and regulatory proteins act as viral transcription factors: nucleoprotein (N), phosphoprotein (P), large polymerase protein (L), and the M2-1 and M2-2 proteins. Created by Biorender.

**Figure 2 viruses-17-00793-f002:**
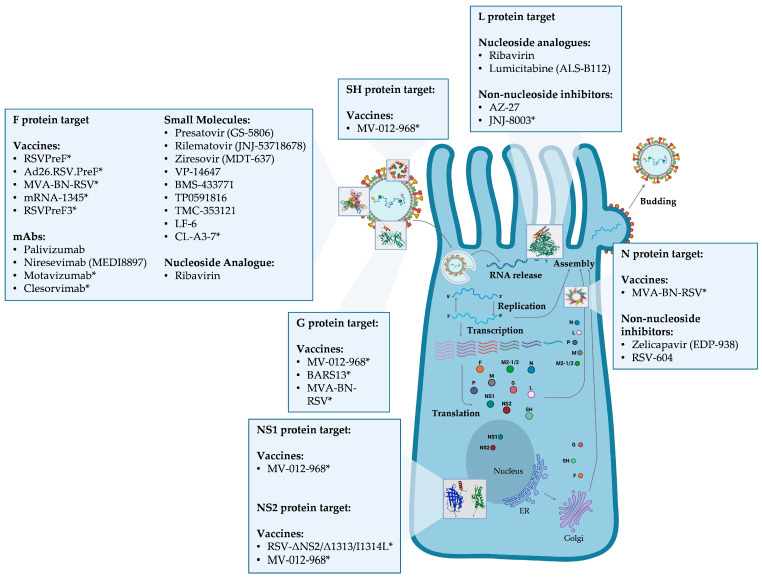
**RSV replication cycle and target sites for preventive and therapeutic strategies.** The figure reports a schematic representation of the RSV replication cycle. The figure also shows an overview of current therapeutics and novel candidates against RSV, including new recombinant antibodies, small molecules such as fusion inhibitors, nucleoprotein inhibitors, nucleoside analogues, and non-nucleoside inhibitors. * Therapeutic strategies not mentioned in this review. F protein (Uniprot P12568), G protein (Uniprot P03423), N protein (Uniprot P03418), L protein (Uniprot P28887), SH protein [72], NS1/NS2 [73]. Created by Biorender.

**Figure 3 viruses-17-00793-f003:**
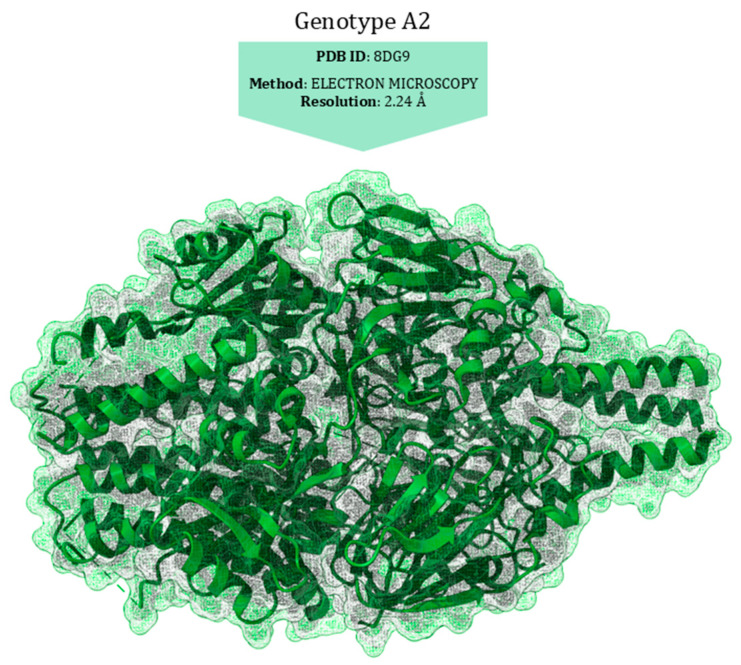
Cryo-EM structure of the RSV pre-fusion F trimer, derived from the model with PDB ID 8DG9 [93].

**Figure 4 viruses-17-00793-f004:**
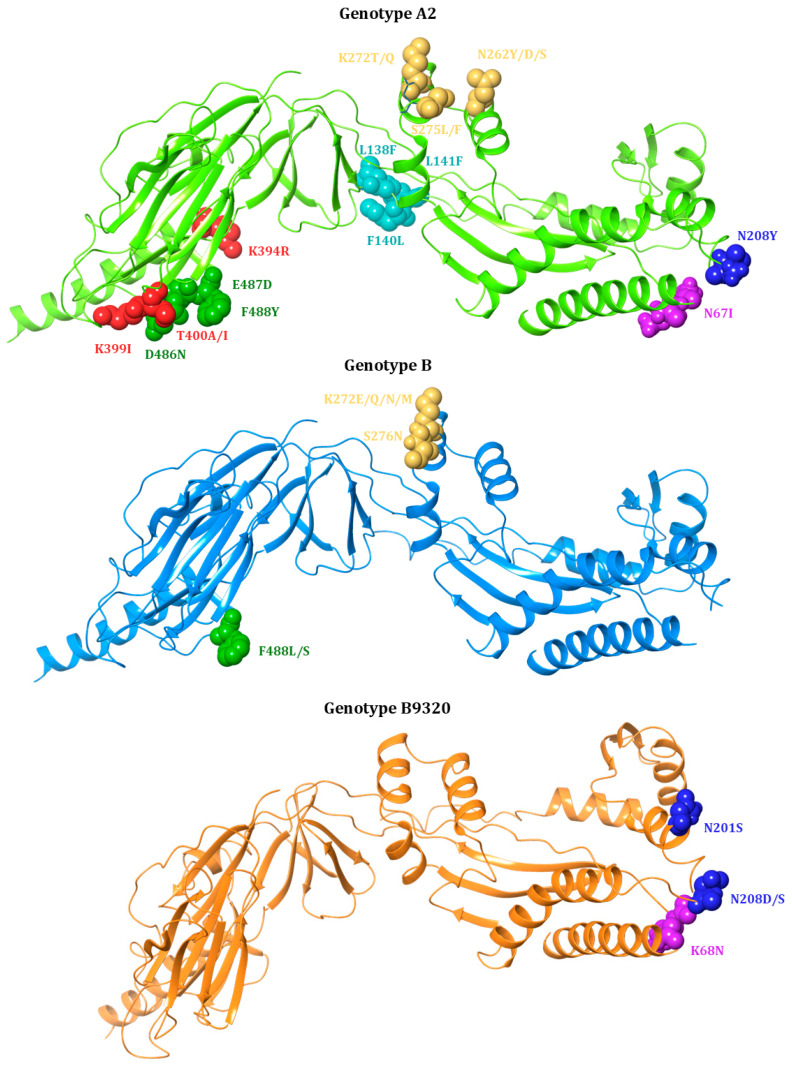
3D monomeric structures of genotype A2, genotype B, and genotype B9320 RSV fusion glycoprotein. In the absence of a resolved 3D structure for genotypes B and B9320, the visualisation of mutations was performed using the PDB-deposited model of genotype A (PDB ID: 8DG9) as a template [79]. Mutated residues are represented in Corey-Pauling-Koltun (CPK); specifically, substitutions in antigenic site A, in F1 domain, in F2 domain, in fusion peptide, in cysteine-rich domain, and in heptad repeat domain are highlighted in yellow, blue, violet, light-blue, red, and green, respectively.

**Figure 5 viruses-17-00793-f005:**
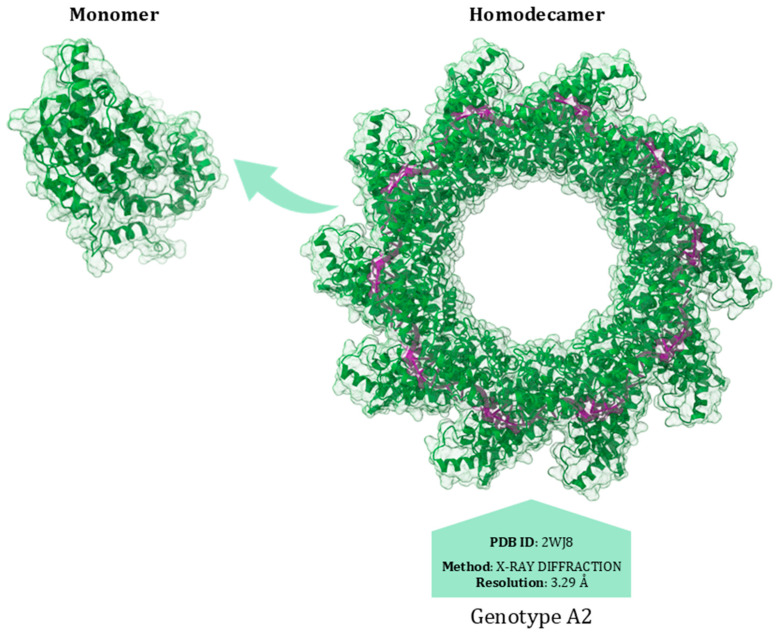
The homodecameric structure, wrapped by the RNA belt, and monomeric structure of the RSV nucleoprotein, derived from the crystallographic model with PDB ID 2WJ8 [126].

**Figure 6 viruses-17-00793-f006:**
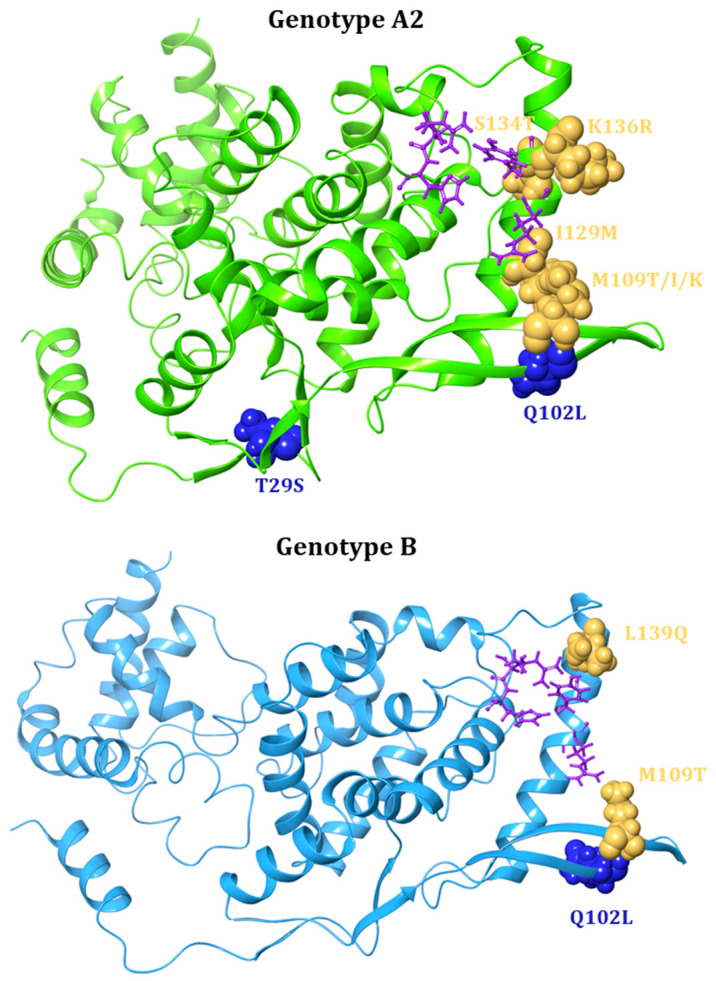
3D monomeric structures of genotype A2 and genotype B RSV nucleoprotein. In the absence of a resolved 3D structure for genotypes B, the visualisation of mutations was performed using the PDB-deposited model of genotype A (PDB ID: 2WJ8) as a template [101]. Key amino acid residues for interaction with the phosphoprotein are shown in purple ball-and-sticks. Mutated residues are represented in Corey-Pauling-Koltun (CPK); specifically, those near to the N/P interaction site are shown in yellow, and those more than 10 Angstroms away are shown in blue.

**Figure 7 viruses-17-00793-f007:**
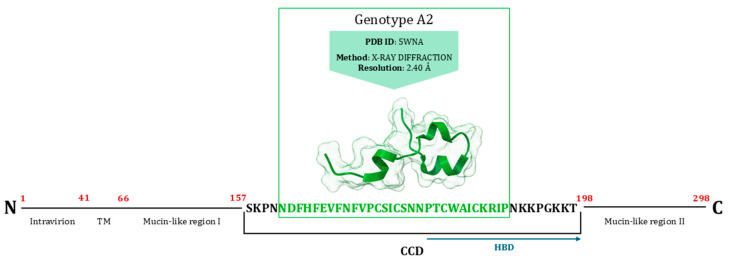
A schematic representation of the RSV G glycoprotein from RSV strain A2. The green residues of the CCD have been resolved in the crystallographic structure with PDB ID 5WNA [109].

**Figure 8 viruses-17-00793-f008:**
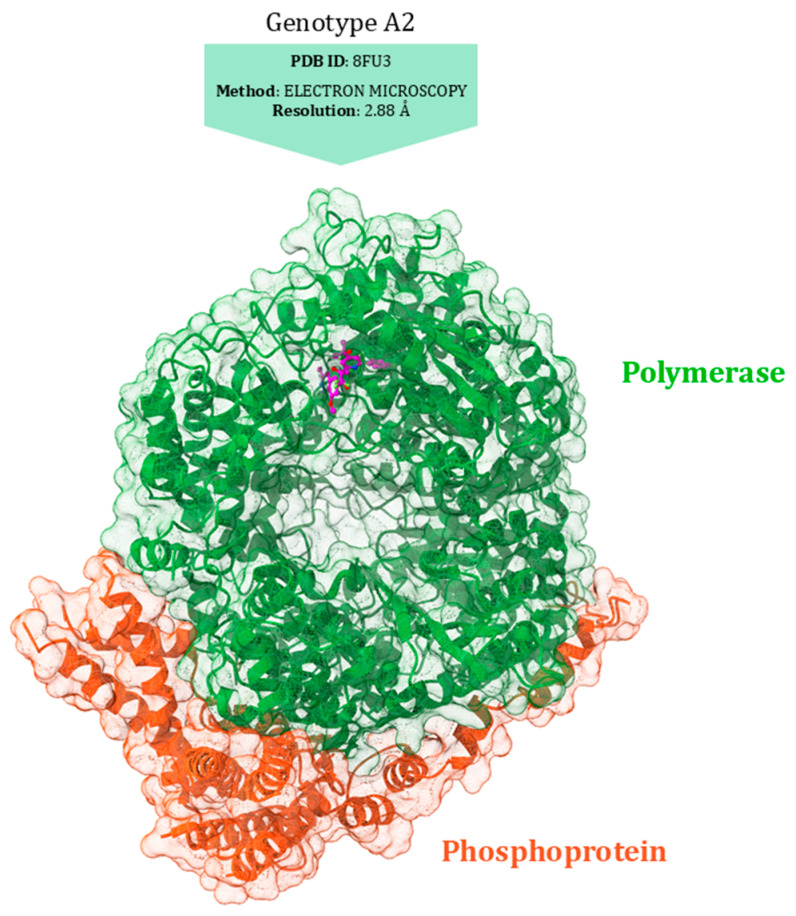
Structure of the RSV polymerase complex (L and P) with a novel non-nucleoside inhibitor (JNJ-8003), derived from the model with PDB ID 8FU3 [144].

**Figure 9 viruses-17-00793-f009:**
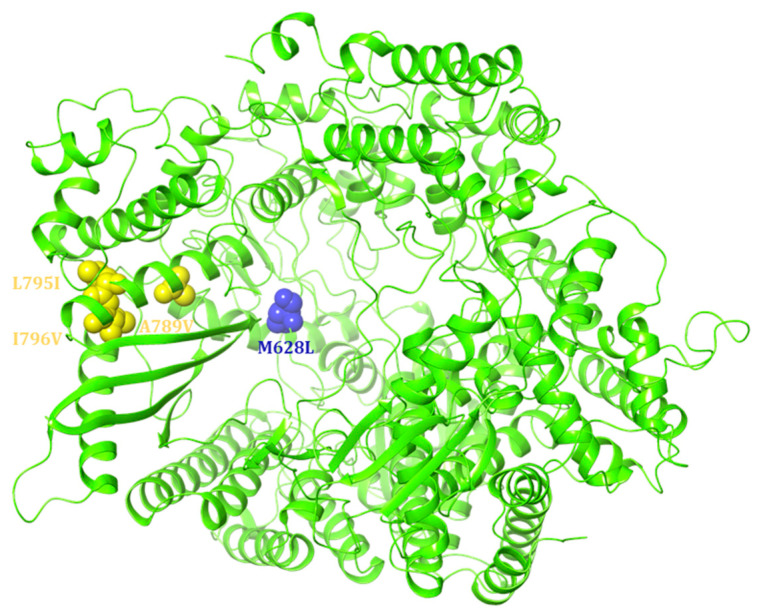
3D structure of genotype A2 RSV polymerase, using the model with PDB ID 8FU3 [106]. Mutated residues are represented in Corey-Pauling-Koltun (CPK); specifically, substitutions in motif B are highlighted in yellow, while the M628L substitution in motif F is highlighted in blue.

**Table 2 viruses-17-00793-t002:** DisoMine analysis of the genotype A2 RSV fusion glycoprotein.

Genotype A2		
AA	Backbone Dynamics	Δ WT/Mutant
N208	0.863	
*Y208*	0.938	+75
N67	0.812	
*I67*	0.892	+80
N262	0.852	
*Y262*	0.937	+85
*D262*	0.888	+36
K272	0.814	
*T272*	0.880	+66
*Q272*	0.825	+11
S275	0.807	
*L275*	0.896	+89
*F275*	0.897	+90
L138	0.896	
*F138*	0.909	+13
F140	0.881	
*L140*	0.888	+7
L141	0.864	
*F141*	0.873	+9
K394	0.920	
*R394*	0.957	+37
K399	0.826	
*I399*	0.881	+55
T400	0.832	
*A400*	0.878	+46
*I400*	0.878	+46
D486	0.804	
*N486*	0.781	−23
E487	0.812	
*D487*	0.803	−9
F488	0.827	
*Y488*	0.820	−7
N517	0.794	
*I517*	0.874	+80

**Table 3 viruses-17-00793-t003:** DisoMine analysis of the genotype B RSV fusion glycoprotein.

**Genotype B**		
**AA**	**Backbone Dynamics**	**Δ WT/Mutant**
K272	0.802	
*E272*	0.813	+11
*Q272*	0.799	−3
*N272*	0.778	−24
*M272*	0.818	+16
S276	0.797	
*N276*	0.807	+10
F488	0.827	
*L488*	0.809	−18
*S488*	0.738	−89

**Table 4 viruses-17-00793-t004:** DisoMine analysis of the genotype B9320 RSV fusion glycoprotein.

Genotype B9320		
AA	Backbone Dynamics	Δ WT/Mutant
K68	0.821	
*N68*	0.797	−24
N201	0.876	
*S201*	0.861	−25
N208	0.872	
*D208*	0.896	+24
*S208*	0.863	−9

**Table 5 viruses-17-00793-t005:** DisoMine analysis of the genotype A2 RSV nucleoprotein.

Genotype A2		
AA	Backbone Dynamics	Δ WT/Mutant
T29	0.666	
*S29*	0.625	−41
S134	0.816	
*T134*	0.857	+41
M109	0.844	
*T109*	0.837	−7
*I109*	0.883	+39
*K109*	0.827	−17
Q102	0.830	
*L102*	0.873	+43
I129	0.868	
*M129*	0.828	−40
K136	0.818	
*R136*	0.822	+4

**Table 6 viruses-17-00793-t006:** DisoMine analysis of the genotype B RSV nucleoprotein.

Genotype B		
AA	Backbone Dynamics	Δ WT/Mutant
M109	0.849	
*T109*	0.843	−6
L139	0.792	
*Q139*	0.794	+2

**Table 7 viruses-17-00793-t007:** DisoMine analysis of the genotype A2 RSV G glycoprotein.

Genotype A2		
AA	Backbone Dynamics	Δ WT/Mutant
K205	0.671	
*G205*	0.614	−57
R8	0.756	
*H8*	0.748	−8
K213	0.635	
*G213*	0.586	−49
T219	0.627	
*A219*	0.626	−1

An increase in backbone flexibility was observed in all mutations.

**Table 8 viruses-17-00793-t008:** DisoMine analysis of the genotype A2 RSV polymerase.

Genotype A2		
AA	Backbone Dynamics	Δ WT/Mutant
A789	1017	
*V789*	1053	+36
M628	0.819	
*L628*	0.842	+23
I796	0.959	
*V796*	0.980	+21
L795	0.989	
*I795*	1008	+19
Y1631	0.914	
*H1631*	0.859	−55

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
