# Peer review of "Insights into the Currently Available Drugs and Investigational Compounds Against RSV with a Focus on Their Drug-Resistance Profiles"

_viruses, 2025, doi:10.3390/v17060793_

Round 1
Reviewer 1 Report
Comments and Suggestions for Authors
This review addresses the need to have a summary of currently available therapeutics for RSV-caused disease. The authors list and discuss a broad range of therapeutic approaches and drugs or drug candidates, including resistance-associated mutations. While the latter aspect is of high interest to the reader, the document falls short of establishing (at least in a speculative or structure-based way) a possible link between mutations and the proposed mode of action of the respective drugs (predicting emergence of resistance or assisting a way how to circumvent this).
Especially the tables 2-4 and later ones presenting DisoMine results remain largely without a link to the text: To which extend will an "increased rigidity" predict (or not) an emerging drug resistance? Obviously both higher rigidity as well as higher flexibility may be linked to lower drug susceptibility. - This clearly requires a central need for more explanation in the text (or these tables should be eliminated, of no general rule can be deduced and explained!)
Other questions that require to be addressed:
Figure 2 legend: please delete the lengthy scientific information from the legend. It should not describe viral processes but only information shown in the graph.
(Fig.4)- Genotype B9320 appears structurally quite distinct - for the more distant reader: what is the basis for this and what are therapeutic consequences in vivo? Please expand on this as well on possible roles of mutations 68N and 201S, which seem to be only relevant for this genotype(?);
lines 361-365: do 488Y and 400I only play a role for the other genotypes?
lines 382/3: does this have implications ALSO IN VIVO?
line 433: what does "partial but noticeable reduction" mean? Does this have implications in vivo or not? please specify.
lines 409, 436: correct "hiper..." to "hypersusceptibility" - what could this mean for the clinical situation? Does this have therapeutic consequences?
line 452: what do you mean by "also" here?
line 466-473: you will need to explain in much more detail why you propose DisoMine here! Also, please elaborate on the predictive nature of higher rigidity or higher flexibility (if there is such a clear rule!)
line 543: it is not the drugs that exhibit resistance mutations, it is the virus! please rephrase the sentence!
line 617: please detail, WHY you find this Interesting.
line 623/4: you have already described Zelicapavir in the text above as "inhibitor of the N protein" - this sentence is not needed here.
lines 740-744 are no "conclusions" - please delete.
lines 744-749 are re-iterations of the text before - it would be helpful to find true conclusions and forward-looking statements in this section.
The very comprehensive bibliography lists 136 references. Among them are a number of quite old publications. Please re-consider eliminating those describing drug candidates that stem from years before 2010 (most of them will either have newer clinical data, or they are of no scientific bearing anymore in 2025.
The text should be re-read by a native English speaker to improve grammar and eliminate misnomers.
change some phrasing, e.g. lines 206: "cell membranes" (which ones (plural here?)); li 208 "shape adjustment" - you may rather mean "conformational change" here(?); li 221: ER matures into Golgi, therefore a "carrying" of proteins to the Golgi is not correct; li 226: "syncytial" are the true translation of cell fusion events (= then "cell syncytia" is a word doubling)...
Comments on the Quality of English LanguageGrammar and some scientific phrasing have to be improved
Author Response
Answer to Reviewer 1 comments
- This review addresses the need to have a summary of currently available therapeutics for RSV-caused disease. The authors list and discuss a broad range of therapeutic approaches and drugs or drug candidates, including resistance-associated mutations. While the latter aspect is of high interest to the reader, the document falls short of establishing (at least in a speculative or structure-based way) a possible link between mutations and the proposed mode of action of the respective drugs (predicting emergence of resistance or assisting a way how to circumvent this). Especially the tables 2-4 and later ones presenting DisoMine results remain largely without a link to the text: To which extend will an "increased rigidity" predict (or not) an emerging drug resistance? Obviously both higher rigidity as well as higher flexibility may be linked to lower drug susceptibility. - This clearly requires a central need for more explanation in the text (or these tables should be eliminated, of no general rule can be deduced and explained!)
Answer:
We would like to thank the reviewer for her/his viewpoint, which we fully agree with. Indeed, increased rigidity or flexibility does not necessarily correlate with enhanced resistance to a given drug. It is true that a specific mutation at an amino acid residue alters the rigidity and flexibility of the surrounding protein environment and (especially when the number of mutations is high) of the entire protein. Of course, an increase in rigidity or flexibility at a specific amino acid position, mainly if located within the binding site, may affect the binding ability of a specific modulator to that site. We aimed to emphasize how a specific resistance mutation can alter the overall behavior of aminoacidic residues in terms of rigidity and flexibility and how this aspect may, in turn, influence drug binding and induced fit to a specific protein target. Since this is a review, our investigation has been rather preliminary; however, our goal is to encourage further exploration of this aspect in the future using more accurate and specific techniques, such as molecular dynamics simulations of WT-drug and mutant-drug complexes. In summary, an amino acid mutation within a specific binding site alters the binding affinity between an inhibitor and that site due to changes in its dynamic behavior. However, based on your recommendation, we aim to explain how a mutation affecting the flexibility of a given residue contributes to modifications in the binding affinity of a specific drug, whether by increasing rigidity or enhancing flexibility. Therefore, according to the Reviewer’s considerations, we modified paragraph 6.4 “Structural Characterization of Drug-Resistance Mutations” and the description of the structural analysis throughout the revised version of the manuscript (page 15, Lines 474-513).
Moreover, for each analyzed target, we reported DisoMine analysis only of the backbone since it gives an unequivocal measure of the intrinsic rigidity of the protein.
Therefore, we modified all the tables and the comments regarding the different tables.
- Figure 2 legend: please delete the lengthy scientific information from the legend. It should not describe viral processes but only information shown in the graph.
Answer:
We thank the Reviewer for his/her suggestion, the legend has been shortened accordingly.
- Fig.4: Genotype B9320 appears structurally quite distinct - for the more distant reader: what is the basis for this and what are therapeutic consequences in vivo? Please expand on this as well on possible roles of mutations 68N and 201S, which seem to be only relevant for this genotype(?);
Answer:
We thank the Reviewer for his/her comment. The RSV strain B9320 is widely used in in vitro experiments as a prototype B genotype.
As the Reviewer pointed out, the mutations 68N and 201S seem to play a role in Nirsevimab resistance mainly in genotype B. Indeed, a previous study, based on in vitro selection experiments, has shown the emergence of these mutations only in the RSV strain B9320 under the selective pressure imposed by Nirsevimab (Zhu et al., J Inf Dis 2018). This is an interesting finding highlighting that the genetic backbone of the different genotypes may predispose to the generation of genotype-dependent genetic profiles associated with drug-resistance. This point has been discussed in the revised version of the manuscript (page 12, lines 370-375).
- lines 361-365: do 488Y and 400I only play a role for the other genotypes?
Answer:
Both mutations have been described in genotype A2, thus their role in other genotypes has not been elucidated yet.
- lines 382/3: does this have implications ALSO IN VIVO?
Answer:
We thank the Reviewer for this comment. As highlighted by previous studies, mutations conferring resistance to TMC-353121 are detrimental for viral replication. It is plausible to hypothesize that viral strains carrying such mutations can rarely emerge in vivo as a consequence their impaired viral replicative capacity. This concept has been reported in the revised version of the manuscript (page 13, lines 393-394).
- line 433: what does "partial but noticeable reduction" mean? Does this have implications in vivo or not? please specify.
Answer:
We agree with the Reviewer’s comment. In order to improve the readability and the interpretation of FC values throughout the manuscript, we provide a definition of high, intermediate and low level resistance in the paragraph 5 entitled “RSV Proteins as Targets of RSV Inhibitors and Related Drug Resistance Mutations” (page 6, lines 236-238). In particular, FC values > 1000 were considered to confer high level of resistance to treatment, FC values ranging from 1000 to 100 intermediate resistance and FC < 100 low resistance. According to these definitions the sentence has been changed as follows: Substitutions like E487D in the heptad repeat region and K399I in the cysteine rich region of the F protein (RSV-A2 strain) confer a lower level of resistance with a FC value of 75 and 45, respectively, indicating a partial reduction in the drug’s efficacy (page 14, lines 442-445).
- lines 409, 436: correct "hiper..." to "hypersusceptibility" - what could this mean for the clinical situation? Does this have therapeutic consequences?
Answer:
We apologize for the typo that has been corrected. According with the Reviewer’s comment, we provide the definition and explanation for mutations associated with hypersusceptibility in the paragraph 5 entiled “RSV Proteins as Targets of RSV Inhibitors and Related Drug Resistance Mutations” (page 6, lines 238-240).
- line 452: what do you mean by "also" here?
Answer:
According with the Reviewer’s comment, we have remodulated the sentences related to the capability of K394R to confer cross-resistance. In particular, a previous study showed that the presence of K394R alone or in combination with D486N or D489Y can enhance the membrane fusion activity of the F protein, as well as decrease the proportion of prefusion F on the cell surface, thus resulting in a shorter time window for inhibitors to bind (Tang et al., J Virol 2021). This can represent a novel mechanism conferring cross-resistance to multiple fusion inhibitors. This has been included in the revised version of the manuscript (page 15, lines 466-472).
- line 466-473: you will need to explain in much more detail why you propose DisoMine here!
Also, please elaborate on the predictive nature of higher rigidity or higher flexibility (if there is such a clear rule!)
Answer:
We agree with the Reviewer’s comment, and thus we have modified the discussion section accordingly (page 25, lines 731-768).
- line 543: it is not the drugs that exhibit resistance mutations, it is the virus! please rephrase the sentence!
Answer:
We thank the Reviewer for pointing it out; we have rephrased the sentence as follows: Substitutions in the Q102-L139 region confer similar degree of resistance to Zelicapavir and RSV-604. These drugs are characterized by a similar chemotype and bind to the N central core, thus affecting N-N and N-viral RNA interactions (page 18, lines 544-546).
- line 617: please detail, WHY you find this Interesting.
Answer:
According to the current literature, mutations in the G protein can increase the degree of resistance when present along with mutations in the N protein, thus acting as secondary mutations. As recently highlighted for other viruses such as HIV (Hikichi et al, CROI 2025), this finding highlights that the genetic profiles underlying resistance to a specific drug class can be more complex than currently thought, thus involving the interaction among multiple viral proteins. This has been reported in the revised version of the manuscript (page 21, lines 623-630).
- line 623/4: you have already described Zelicapavir in the text above as "inhibitor of the N protein" - this sentence is not needed here.
Answer:
We have described Zelicapavir also in this paragraph in order to highlight that the genetic profiles conferring resistance to this drug can emerge in viral proteins other than the N protein. Further studies are necessary to better clarify the mechanisms underlying this phenomenon.
- lines 740-744 are no "conclusions" - please delete.
Answer:
We have deleted this part accordingly.
- lines 744-749 are re-iterations of the text before - it would be helpful to find true conclusions and forward-looking statements in this section.
Answer:
According with the Reviewer’s comment, we have revised the Conclusion section. In particular, lines 744–749 have been remodulated to avoid repetition, and we have added clearer conclusions along with forward-looking statements, as suggested.
- The very comprehensive bibliography lists 136 references. Among them are a number of quite old publications. Please re-consider eliminating those describing drug candidates that stem from years before 2010 (most of them will either have newer clinical data, or they are of no scientific bearing anymore in 2025.
Answer:
We thank the Reviewer for this remark; references have been updated following his/her suggestion.
- change some phrasing, e.g. lines 206: "cell membranes" (which ones (plural here?)); li 208 "shape adjustment" - you may rather mean "conformational change" here(?); li 221: ER matures into Golgi, therefore a "carrying" of proteins to the Golgi is not correct; li 226: "syncytial" are the true translation of cell fusion events (= then "cell syncytia" is a word doubling)...
Answer:
We thank the Reviewer for his/her suggestion. Sentences have rephrased accordingly (page 6, line 203; page 6, line 216; page 6, line 222).

Reviewer 2 Report
Comments and Suggestions for Authors
Thanks for the invitation to review the current manuscript.
The submitted review by Alessia Magnapera et al., examined the currently available and investigational drugs against RSV. The review is timely and relevant for several reasons including the global health and economic impact of RSV infections and its huge burden especially in pediatric and elderly demographics, the recent advances in RSV therapeutics, and the reported emerging resistance to RSV therapeutics.
The manuscript is well written and fits the Special Issue “Antiviral Resistance Mutations”; however, reformatting is recommended to enhance readability by addition of a separate Methods and Discussion section. Additionally, the implications of review findings and the limitations of the review should be clearly stated.
Based on this, I recommend a major revision to address the following points that hopefully would help the authors to improve their final manuscript:
- In the Abstract, page 1, lines 15 – 16: It is worth mentioning that RSV severe morbidity also involves the elderly as recently appreciated in various studies and reviews.
- In the Introduction, page 1, line 39: Since RSV disproportionately affect infants, it would be more scientifically proper to specify this in the sentence rather than stating that it causes significant children mortality
- In the Introduction, page 1, lines 39 – 41: It is recommended to specify the burden of RSV disease in numbers citing a more recent reference. A suggested revision can be as follows: In 2019, RSV-related lower respiratory tract infections accounted for 338,495 deaths globally, particularly among children and older adults. A suggested citation: Wu M, Wu Q, Liu D, Zu W, Zhang D, Chen L. The global burden of lower respiratory infections attributable to respiratory syncytial virus in 204 countries and territories, 1990-2019: findings from the Global Burden of Disease Study 2019. Intern Emerg Med. 2024 Jan;19(1):59-70. doi: 10.1007/s11739-023-03438-x. Epub 2023 Oct 3. PMID: 37789183.
- In the Introduction, it would be suitable to provide a short overview on the suitability of F-protein as a target for therapeutic and preventive measures against RSV.
- My major comment is the need to add a methods section to detail how the review was conducted and the exact basis for literature search in terms of the databases covered, keywords or MeSH terms used for the search, the date of search concluded, and inclusion/exclusion criteria if applicable. This is needed to enable other authors to replicate the findings and to assess the limitations of the review.
- Another major comment is the need for a Discussion section to contextualize the findings of the review and DisoMine analysis and its potential clinical implications. Additionally, the Discussion section needed should include the limitations of the review and analysis.
- Line 154: regarding the role of RSV SH Protein, please check a recently published article: Okura T, Takahashi T, Kameya T, Mizukoshi F, Nakai Y, Kakizaki M, Nishi M, Otsuki N, Kimura H, Miyakawa K, Shirato K, Kamitani W, Ryo A. MARCH8 Restricts RSV Replication by Promoting Cellular Apoptosis Through Ubiquitin-Mediated Proteolysis of Viral SH Protein. Viruses. 2024 Dec 18;16(12):1935. doi: 10.3390/v16121935. PMID: 39772241; PMCID: PMC11680241.
- In section 3 “RSV Virion and Genome”, page 4, lines 156 – 164: It is recommended that the authors provide a more in-depth description of the distinct yet complementary roles of RSV F and G proteins in viral entry and immune evasion.
- Lines 252 – 253: It better to move this sentence into a separate Methods section.
- Lines 292 – 294: A reference is needed to support this statement regarding the protection provided by Palivizumab prophylaxis. A suggested recent reference: O'Hagan S, Galway N, Shields MD, Mallett P, Groves HE. Review of the Safety, Efficacy and Tolerability of Palivizumab in the Prevention of Severe Respiratory Syncytial Virus (RSV) Disease. Drug Healthc Patient Saf. 2023 Sep 11;15:103-112. doi: 10.2147/DHPS.S348727. PMID: 37720805; PMCID: PMC10503506.
- Line 301. It is recommended that the authors elaborate on RSV amino acid substitutions before Table 1.
- Line 303: What is the Enzimatic assay? I think it a typographical error. Please check
- Lines 466 – 468: It is better to be moved into a separate Methods section.
- It is better to use the abbreviation upon the first encounter and to be consistent in terminology. E.g., lower respiratory tract infections (LRTIs): line 35, line 40, line 53; RSV: line 33, line 87, 734; monoclonal antibodies (mAbs): lines 281, 285, 301, 603, 744
- In Figure 2, the authors are recommended to increase the font size to enhance the readability and to use the MDPI recommended font in all figures (Palatino Linotype)
- In Table 1 and in order to render the table standing alone, please spell out all the abbreviations (FC, AA, mAbs) and to remove redundant mentioning of the references in each cell.
- In Table 1. Were the embedded figures in the table created by the authors? If not, please cite the sources of these images.
- In Table 1 and elsewhere in the manuscript where the authors are referring to amino acids (e.g., lines 306, 317, 318, 324, 337, 340, 342, 345, 352, 354, 359, 368, 380, 382, 384, 386, 402, 408, 419, 435, 446, 455), it would be more accurate the replace the term mutation with substitution.
- Figure 4: Please cite the source that was used to create the figure. The same applies to Figures 6 and 9.
- Lines 370 – 372: It is better to have this as a major heading followed by subheadings in the sections below.
- Line 488 (Table 2). I suggest reformatting the table to enable easier reading without the need for colors (e.g., by adding a new columns for description of dynamics based on DisoMine analysis. The same applies to Tables 3, 4, 5, 6, 7 and 8.
Best wishes!
Author Response
Answer to Reviewer 2 comments
- The submitted review by Alessia Magnapera et al., examined the currently available and investigational drugs against RSV. The review is timely and relevant for several reasons including the global health and economic impact of RSV infections and its huge burden especially in pediatric and elderly demographics, the recent advances in RSV therapeutics, and the reported emerging resistance to RSV therapeutics.
The manuscript is well written and fits the Special Issue “Antiviral Resistance Mutations”;
Answer:
We thank the Reviewer for his/her positive comment.
- Reformatting is recommended to enhance readability by addition of a separate Methods and Discussion section. Additionally, the implications of review findings and the limitations of the review should be clearly stated. Based on this, I recommend a major revision to address the following points that hopefully would help the authors to improve their final manuscript.
Answer:
The review has been largely reformatted based on the Reviewer’s suggestions. A separate Methods section has been included as supplementary text. Furthermore, we have included a Discussion section to address the implications and the limitations of the review findings (page 25, lines 731-768).
- In the Abstract, page 1, lines 15 – 16: It is worth mentioning that RSV severe morbidity also involves the elderly as recently appreciated in various studies and reviews.
Answer:
A sentence reporting RSV severe morbidity in elderlies has been included in the Abstract accordingly (page 1, line 15).
- In the Introduction, page 1, line 39: Since RSV disproportionately affect infants, it would be more scientifically proper to specify this in the sentence rather than stating that it causes significant children mortality
Answer:
We modified the sentence in the introduction accordingly.
- In the Introduction, page 1, lines 39 – 41: It is recommended to specify the burden of RSV disease in numbers citing a more recent reference. A suggested revision can be as follows: In 2019, RSV-related lower respiratory tract infections accounted for 338,495 deaths globally, particularly among children and older adults. A suggested citation: Wu M, Wu Q, Liu D, Zu W, Zhang D, Chen L. The global burden of lower respiratory infections attributable to respiratory syncytial virus in 204 countries and territories, 1990-2019: findings from the Global Burden of Disease Study 2019. Intern Emerg Med. 2024 Jan;19(1):59-70. doi: 10.1007/s11739-023-03438-x. Epub 2023 Oct 3. PMID: 37789183.
Answer:
We have appreciated the Reviewer’s comment. The citation and the reference have been included in the revised version of the manuscript (page 1, lines 41-42).
- In the Introduction, it would be suitable to provide a short overview on the suitability of F-protein as a target for therapeutic and preventive measures against RSV.
Answer:
We thank the Reviewer for his/her suggestions. We have better specified the importance of F protein as a therapeutic target in the Introduction of the revised version of the manuscript (page 2, lines 52-54). The role of the protein is then addressed more specifically in the paragraph 6.
- My major comment is the need to add a methods section to detail how the review was conducted and the exact basis for literature search in terms of the databases covered, keywords or MeSH terms used for the search, the date of search concluded, and inclusion/exclusion criteria if applicable. This is needed to enable other authors to replicate the findings and to assess the limitations of the review.
Answer:
According to the Reviewer’s comment, we have included a Methods Section as Supporting Information.
- Another major comment is the need for a Discussion section to contextualize the findings of the review and DisoMine analysis and its potential clinical implications. Additionally, the Discussion section needed should include the limitations of the review and analysis.
Answer:
According with the Reviewer’s comment, we have included a Discussion Section in which the potential clinical implications have been included (page 25, lines 731-768).
- Line 154: regarding the role of RSV SH Protein, please check a recently published article: Okura T, Takahashi T, Kameya T, Mizukoshi F, Nakai Y, Kakizaki M, Nishi M, Otsuki N, Kimura H, Miyakawa K, Shirato K, Kamitani W, Ryo A. MARCH8 Restricts RSV Replication by Promoting Cellular Apoptosis Through Ubiquitin-Mediated Proteolysis of Viral SH Protein. Viruses. 2024 Dec 18;16(12):1935. doi: 10.3390/v16121935. PMID: 39772241; PMCID: PMC11680241.
Answer:
We have appreciated the suggestion and we included this reference in the paragraph accordingly (page 4, lines 157-158).
- In section 3 “RSV Virion and Genome”, page 4, lines 156 – 164: It is recommended that the authors provide a more in-depth description of the distinct yet complementary roles of RSV F and G proteins in viral entry and immune evasion.
Answer:
The role of F and G proteins in viral entry and immune evasion has been extensively described in the paragraphs 6 and 8, respectively.
- Lines 252 – 253: It better to move this sentence into a separate Methods section.
Answer:
We have moved the sentence in the Methods Section accordingly.
- Lines 292 – 294: A reference is needed to support this statement regarding the protection provided by Palivizumab prophylaxis. A suggested recent reference: O'Hagan S, Galway N, Shields MD, Mallett P, Groves HE. Review of the Safety, Efficacy and Tolerability of Palivizumab in the Prevention of Severe Respiratory Syncytial Virus (RSV) Disease. Drug Healthc Patient Saf. 2023 Sep 11;15:103-112. doi: 10.2147/DHPS.S348727. PMID: 37720805; PMCID: PMC10503506.
Answer:
We have appreciated this suggestion and we included this reference in the paragraph accordingly.
- Line 301. It is recommended that the authors elaborate on RSV amino acid substitutions before Table 1.
Answer:
According with the Reviewer’s suggestion. we have added a sentence to introduce Table 1 in the paragraph 5 “RSV Proteins as Targets of RSV Inhibitors and Related Drug Resistance Mutations” (page 7, lines 245-247).
- Line 303: What is the Enzimatic assay? I think it a typographical error. Please check
Answer:
We apologize for the typo that has been corrected accordingly.
- Lines 466 – 468: It is better to be moved into a separate Methods section.
Answer:
We have moved the sentence in the Methods Section accordingly.
- It is better to use the abbreviation upon the first encounter and to be consistent in terminology. E.g., lower respiratory tract infections (LRTIs): line 35, line 40, line 53; RSV: line 33, line 87, 734; monoclonal antibodies (mAbs): lines 281, 285, 301, 603, 744
Answer:
According with Reviewer’s comment, we have checked the abbreviation throughout the revised version of the manuscript.
- In Figure 2, the authors are recommended to increase the font size to enhance the readability and to use the MDPI recommended font in all figures (Palatino Linotype)
Answer:
We changed the figures according to suggestions.
- In Table 1 and in order to render the table standing alone, please spell out all the abbreviations (FC, AA, mAbs) and to remove redundant mentioning of the references in each cell.
Answer:
We have modified Table 1 according with the Reviewer’s comment.
- In Table 1. Were the embedded figures in the table created by the authors? If not, please cite the sources of these images.
Answer:
Figures in Table 1 were generated by the authors.
- In Table 1 and elsewhere in the manuscript where the authors are referring to amino acids (e.g., lines 306, 317, 318, 324, 337, 340, 342, 345, 352, 354, 359, 368, 380, 382, 384, 386, 402, 408, 419, 435, 446, 455), it would be more accurate the replace the term mutation with substitution.
Answer:
We have changed the term “mutation” with amino acid substitution throughout the manuscript.
- Figure 4: Please cite the source that was used to create the figure. The same applies to Figures 6 and 9.
Answer:
According to the Reviewer’s comment, we have added the source used to create figures 4, 6 and 9 in the figure legends.
- Lines 370 – 372: It is better to have this as a major heading followed by subheadings in the sections below.
Answer:
We thank the Reviewer for his/her comment that has been addressed accordingly.
- Line 488 (Table 2). I suggest reformatting the table to enable easier reading without the need for colors (e.g., by adding a new columns for description of dynamics based on DisoMine analysis. The same applies to Tables 3, 4, 5, 6, 7 and 8.
Answer:
Following the Reviewer's suggestion, the tables have been reformatted to enhance readability and eliminate the use of colours. Additionally, the side chain dynamics' column has been removed, retaining only the 'backbone dynamics' column, and a new column has been added to report the Δ between WT and mutant with respect to backbone dynamics.

Round 2
Reviewer 1 Report
Comments and Suggestions for Authors
This review of drugs, candidates or new research approaches addressing RSV therapy is provided. This revised version represents an improved manuscript, in which the reviewer's suggestions for correction and change have appropriately been incorporated in the modified manuscript.
The additional information and explanations, as requested or suggested. have now been provided where necessary.
The language use has been improved and is now overall acceptable
Author Response
Answer to Reviewer 1 comments
This review of drugs, candidates or new research approaches addressing RSV therapy is provided.
This revised version represents an improved manuscript, in which the reviewer's suggestions for
correction and change have appropriately been incorporated in the modified manuscript.
The additional information and explanations, as requested or suggested. have now been provided
where necessary. The language use has been improved and is now overall acceptable.
Answer:
We sincerely thank the Reviewer for the positive evaluation of our revised manuscript. We appreciate
your recognition of the improvements made, particularly regarding the incorporation of your valuable
suggestions, the addition of the requested information, and the enhanced language clarity. Your
constructive feedback has been fundamental in strengthening the quality of our work.

Reviewer 2 Report
Comments and Suggestions for Authors
Thanks for the invitation to review the revised version of the manuscript.
Although the authors addressed the majority of minor comments, there is a large room for improvement which are essential to make this manuscript fit into the quality criteria of being accepted for publication in viruses journal as follows:
- The authors stated in their reply that “A separate Methods section has been included as supplementary text.” However, the authors should include the Methods section in the main manuscript as it is an essential part of any efforts aiming to replicate the review findings. Additionally, the supplementary section provided did not include essential parts of the Methods including the databases used for the search, the date of search conclusion, how the authors appraised the quality of included records, etc.
- Regarding the Discussion section, in the revised form this section is uninformative at all. As stated by the viruses journal guidelines: “Authors should discuss the results and how they can be interpreted in perspective of previous studies and of the working hypotheses. The findings and their implications should be discussed in the broadest context possible and limitations of the work highlighted. Future research directions may also be mentioned. This section may be combined with Results.” In the current form, the Discussion lacked these aspects which was manifested by lack of citation of any relevant references and what new insights were provided by the current manuscript.
- The response to my previous point “In section 3 “RSV Virion and Genome”, page 4, lines 156 – 164: It is recommended that the authors provide a more in-depth description of the distinct yet complementary roles of RSV F and G proteins in viral entry and immune evasion” was not satisfactory. This indicates that the authors were unwilling to make this important addition to the revised manuscript. If so, the authors should provide scientific justification for this decision.
- I still believe that Figure 2 in the current format and font size it not readable. This figure should be revised to enhance readability.
- In Table 1, the source of embedded figures should be clearly stated in the Table footnote. I understand that the authors replied “Figures in Table 1 were generated by the authors”; however, this response is not sufficient. The authors should mention the exact software used to generate these figures and in what settings.
Author Response
Answer to Reviewer 2 comments
Although the authors addressed the majority of minor comments, there is a large room for improvement which are essential to make this manuscript fit into the quality criteria of being accepted for publication in viruses journal as follows:
- The authors stated in their reply that “A separate Methods section has been included as supplementary text.” However, the authors should include the Methods section in the main manuscript as it is an essential part of any efforts aiming to replicate the review findings. Additionally, the supplementary section provided did not include essential parts of the Methods including the databases used for the search, the date of search conclusion, how the authors appraised the quality of included records, etc.
Answer:
We thank the Reviewer for the relevant suggestion. In accordance with this recommendation, we have moved the Methods section from the supplementary material to the main manuscript, where it is now presented as Section 6.Furthermore, we have expanded this section to include the previously missing methodological details.
- Regarding the Discussion section, in the revised form this section is uninformative at all. As stated by the viruses journal guidelines: “Authors should discuss the results and how they can be interpreted in perspective of previous studies and of the working hypotheses. The findings and their implications should be discussed in the broadest context possible and limitations of the work highlighted. Future research directions may also be mentioned. This section may be combined with Results.” In the current form, the Discussion lacked these aspects which was manifested by lack of citation of any relevant references and what new insights were provided by the current manuscript.
Answer:
We have appreciated the reviewer’s comments regarding the Discussion section, that has been significantly expanded according with the Reviewer’s suggestions. In this updated version, we discuss our findings in the context of previous studies, highlight the implications and limitations of our work, and suggest directions for future research. We have also added relevant references to support the discussion. As requested by the Journal’s guideline, we have retained it as a separate section.
The response to my previous point “In section 3 “RSV Virion and Genome”, page 4, lines 156 – 164: It is recommended that the authors provide a more in-depth description of the distinct yet complementary roles of RSV F and G proteins in viral entry and immune evasion” was not satisfactory. This indicates that the authors were unwilling to make this important addition to the revised manuscript. If so, the authors should provide scientific justification for this decision.
Answer:
We thank the Reviewer for the additional comment. In the indicated paragraph, we have outlined the roles of the RSV F and G proteins in immune evasion and have added further information regarding their involvement in viral entry (page 4, line 167-183). Furthermore, a more detailed discussion of the biological functions of these proteins has been also provided in paragraphs 4 (page 5, lines 204-210), 7 (page 8, lines 319-324) and 9 (page 22, lines 664-666).
- I still believe that Figure 2 in the current format and font size it not readable. This figure should be revised to enhance readability.
Answer:
According with the Reviewer’s suggestion, we have revised Figure 2 to improve its readability by adjusting the layout and increasing the font size.
- In Table 1, the source of embedded figures should be clearly stated in the Table footnote. I understand that the authors replied “Figures in Table 1 were generated by the authors”; however, this response is not sufficient. The authors should mention the exact software used to generate these figures and in what settings.
Answer:
In accordance with the Reviewer’s suggestion, we have updated the legend of Table 1 by indicating the software used to generate the two-dimensional drug structures.

Round 3
Reviewer 2 Report
Comments and Suggestions for Authors
I would like to thank the authors for their efforts in addressing my previous comments, which were intended to improve the clarity and quality of the manuscript.
While some revisions have been made, I believe that both the Methods and Discussion sections still require significant improvement to meet the expected standards for publication in viruses journal. Additionally, the overall organization of the manuscript could benefit from further revisions to enhance readability and flow. Nevertheless, I will defer the final decision to the academic editor to consider.
Best wishes!
Author Response
Answer to Reviewer 2 comments
I would like to thank the authors for their efforts in addressing my previous comments, which were intended to improve the clarity and quality of the manuscript.
While some revisions have been made, I believe that both the Methods and Discussion sections still require significant improvement to meet the expected standards for publication in viruses journal. Additionally, the overall organization of the manuscript could benefit from further revisions to enhance readability and flow. Nevertheless, I will defer the final decision to the academic editor to consider.
Best wishes!
Answer:
We thank the reviewer for their thoughtful comments and for acknowledging our efforts in addressing the previous round of feedback. We appreciate the additional observations regarding the Methods and Discussion sections, as well as the overall organization of the manuscript.
We fully respect the reviewer’s perspective and defer to the academic editor’s judgment regarding any further revisions that may be required to meet the journal’s standards.
We remain grateful for the reviewer’s time and constructive input, which have been valuable in shaping this manuscript.
